# An Overview of Phase-Locked Loop: From Fundamentals to the Frontier

**DOI:** 10.3390/s25185623

**Published:** 2025-09-09

**Authors:** Thi Viet Ha Nguyen, Cong-Kha Pham

**Affiliations:** Department of Computer and Network Engineering, University of Electro-Communications (UEC), 1-5-1 Chofugaoka, Chofu-shi 182-8585, Tokyo, Japan; phamck@uec.ac.jp

**Keywords:** Phase-Locked Loop, frequency synthesis, frequency divider, PLL bandwidth, timing recovery, signal synchronization, high-frequency oscillators, phase noise

## Abstract

Phase-Locked Loops (PLLs) are fundamental building blocks in modern electronic systems, enabling precise frequency synthesis, signal synchronization, and clock generation. This paper provides a comprehensive system-level analysis of PLLs, covering their fundamental operation, key architectures, performance considerations, and applications in emerging technologies. Additionally, this paper discusses current challenges and future trends in PLL design, including low-power optimization, noise reduction, and integration with machine learning techniques.

## 1. Introduction

In the era of burgeoning digital communication, Phase-Locked Loops (PLLs) have emerged as pivotal components in modern electronic systems, enabling precise frequency synthesis and robust signal synchronization essential for high-performance applications. From wireless communication networks and radar systems to automotive electronics, PLLs underpin signal integrity, minimize phase noise, and ensure accurate timing, directly impacting system reliability and efficiency.

The relentless demand for increased data rates, miniaturization, and enhanced energy efficiency has driven significant advancements in PLL architectures. This evolution encompasses a variety of designs, such as Analog PLLs (APLLs), which offer low-jitter performance suited for high-frequency analog systems. In contrast, digital PLLs (DPLLs) and all-digital PLLs (ADPLLs) provide enhanced programmability and integration capabilities in deep-submicron CMOS processes. Additionally, integer-N and fractional-N PLLs differ in frequency resolution and spurious performance, while Injection-Locked PLLs (ILPLLs) and Delay-Locked Loops (DLLs) serve specific roles in low-power or clock distribution applications.

The existing literature on PLL design features several key works that have advanced our understanding of these systems. The recent text by Rhee and Yu (2023) [1] offers a modern, comprehensive treatment, establishing a broad foundation for both the theory and implementation of contemporary PLLs. Focusing more specifically on digital implementations, Rohde et al. (2021) [2] provide a detailed analysis of digital PLL synthesizers, with a particular emphasis on their design for microwave and wireless systems. From an application-centric viewpoint, Farzaneh et al. (2018) [3] explore the role of PLLs within wireless communication circuits, detailing their function in core applications like FM modulation and demodulation. Although these contributions have been instrumental, they tend to offer either a broad overview, concentrate on a specific high-frequency implementation paradigm, or focus on a narrow set of applications. This highlights the need for a comprehensive review that bridges these distinct perspectives, which the present work aims to provide.

Specifically, the rapid emergence of Internet-of-Things (IoT) applications has introduced stringent power consumption requirements that are inadequately addressed by existing surveys, which predominantly focus on performance optimization without systematic consideration of power-performance trade-offs. Furthermore, the proliferation of millimeter-wave (mmWave) communication systems for 5G and beyond has created new challenges in frequency synthesis and phase noise management at unprecedented frequencies—topics that remain underexplored in the current literature. Additionally, recent advances in digital calibration techniques, adaptive bandwidth control, and machine learning-assisted PLL optimization represent significant developments that lack comprehensive treatment in existing reviews.

Compared with previous reviews, this paper contributes in three main ways. First, a classification framework based on the power-performance trade-off is introduced to evaluate PLL architectures specifically designed for emerging low-power applications. Second, this work provides an in-depth analysis of recent design techniques, especially for IoT and mmWave applications, which have not been explored in previous reviews. Third, through a comparative discussion of advanced techniques such as phase noise, jitter, spur, and digital calibration, this paper provides practical insights to support the development of next-generation PLL designs.

To emphasise these contributions, this work is organized into five primary parts:Section 2 reviews the historical evolution of PLLs, emphasizing key innovations.Section 3 provides a system-level analysis, explaining their fundamental principles and advantages.Section 4 addresses major challenges in PLL design and evaluates current solutions.Section 5 surveys recent advancements in PLL architectures and their diverse applications.Section 6 concludes with reflections on the current state of PLL technology and predictions for its future trajectory.

## 2. A Brief History—A Journey Through Time

This section provides more compressed coverage of earlier historical periods (pre-2018) while offering expanded detail for the most recent decade (2018–2025), which reflects both the accelerating pace of innovation in this domain and the particular relevance of recent developments to contemporary system requirements. The period from 2018 onwards has witnessed unprecedented innovation velocity, driven primarily by emerging demands in high-speed communication infrastructures, Internet-of-Things (IoT) applications, and millimeter-wave systems. This temporal emphasis facilitates comprehension of current technological trajectories while contextualizing recent breakthroughs within the broader evolutionary continuum. Presenting these advancements along a year-by-year timeline is intended to help readers, especially those new to the field, grasp the pace of development, observe technological trajectories, and connect innovation milestones with broader system-level trends such as low-jitter design, hybrid analog-digital integration, and energy efficiency.

The Evolution of Phase-Locked Loops: While the Phase-Locked Loop (PLL) concept emerged in the 1930s, it really took off in the 1950s and 1960s [4,5]. During this period, PLLs found their first significant uses in radio signal modulation and demodulation. Early PLL systems were primarily Analog Phase Locked Loop (APLLs) and were integrated into technologies like televisions, radar, and motor control systems.

The 1970s and 1980s marked a turning point with the rapid growth of semiconductor technology and the advent of Integrated Circuits (ICs). This made PLLs indispensable in telecommunications and digital signal processing. They became key components in frequency synthesizers, microcontrollers, and first-generation mobile phones. By the 1990s, the explosive development of CMOS technology ushered in a new era for PLLs, especially in microprocessors and consumer electronics. CMOS greatly boosted PLL performance, slashed power consumption, and allowed direct integration into complex microcircuits. At the same time, digital PLLs (DPLLs) emerged, revolutionizing PLL operation by boosting accuracy and cutting down on phase noise [4,5].

Accelerating Innovation (2000–2017): From the 2000s to 2017, PLL design underwent significant transformation, focusing on improvements in power efficiency, noise performance, and integration. All-digital PLLs (ADPLLs) [6,7] became mainstream, leveraging Time-to-Digital Converters (TDCs) for lower jitter, finer frequency resolution, and reduced size and power. Fractional-N PLLs, coupled with advanced Sigma-Delta Modulation, minimized spurs and phase noise, enhancing their performance in RF and high-speed digital systems [8,9,10,11].

Recent Milestones and Future Trajectories (2018–2025): In 2018, charge pump PLLs (CP-PLLs) saw significant enhancements. Optimizations in charge pump designs led to reduced leakage currents, balanced voltage levels, and improved linearity, resulting in lower phase noise and greater stability [7]. Additionally, the adoption of closed-loop amplifiers in PLLs improved loop response, reduced power consumption, and boosted frequency accuracy [12].

The year 2019 saw a major breakthrough with Injection-Locked PLLs (IL-PLLs). These achieved superior low-jitter performance and rapid locking times by using injection-locking mechanisms to enhance phase noise suppression and frequency stability [13].

In 2020, the cascaded-PLL framework emerged as a powerful solution for high-order noise shaping. By nesting multiple PLLs, it effectively suppressed phase noise across various frequency stages, improving spectral purity and stability. Also gaining traction were Sub-Sampling PLLs (SS-PLLs), which achieved low power consumption with excellent phase noise performance by directly sampling the reference clock, minimizing noise from traditional phase detectors [14].

The year 2021 saw the application of kT/C noise-cancellation techniques, which significantly reduced thermal noise in PLLs, leading to cleaner signal outputs and improved overall performance [15]. Furthermore, digital PLLs (DPLLs) advanced by integrating sophisticated digital signal processing (DSP) techniques, enhancing frequency resolution, stability, and adaptability. The shift toward all-digital architectures offered greater programmability and process scaling, making DPLLs a preferred choice for modern high-speed communication and energy-efficient frequency synthesis solutions [16].

Advanced loop filtering techniques introduced in 2022 optimized loop filter designs, leading to better phase noise characteristics and reduced jitter [17]. Fractional-N PLLs (F-PLLs) continued to evolve, offering finer frequency resolution and improved performance for frequency synthesis applications [18].

High-efficiency amplification solutions emerged in 2023, providing robust performance for PLLs across various applications. Ring Oscillator PLLs (RO-PLLs) saw notable advancements, offering compact designs with low power consumption and fast locking times [19].

In 2024, a non-reductionist modeling approach for PLLs allowed for more realistic simulations and precise definitions of synchronization regimes. Meanwhile, Millimeter-wave PLLs (mm-Wave PLLs) made substantial improvements, enabling high-frequency operation and low phase noise critical for 5G and beyond applications [20].

Looking ahead to 2025, we note that PLLs are set for significant advancements, driven by the increasing digitization of communication systems and the demand for power efficiency. ADPLLs, leveraging Time-to-Digital Converters and Digitally Controlled Oscillators, are expected to dominate low-power applications like Wi-Fi, Bluetooth, and IoT, offering seamless CMOS integration and programmable flexibility [21]. For high-frequency domains, such as 5G mm-Wave, hybrid PLLs will combine analog precision with digital control to achieve low phase noise and high bandwidth [22]. Innovations in delta-sigma modulation and adaptive loop bandwidth will enhance phase noise suppression and energy efficiency, while sub-sampling techniques address challenges in mm-Wave digitization [23]. These advancements will solidify PLLs as critical enablers of scalable, multi-band wireless systems, expertly balancing performance and integration in next-generation transceivers.

From their analog beginnings in the 1930s to the sophisticated architectures of 2025, the evolution of PLLs underscores their vital role in electronics. The shift to all-digital designs, coupled with innovations like fractional-N, Injection-Locked, Sub-Sampling, and DLLs, has dramatically improved power efficiency, phase noise performance, and integration [8,9,14,22]. Advances in CMOS technology, along with techniques like kT/C noise cancellation and the potential for machine learning optimization, have boosted their precision and versatility [20,21]. As communication systems continue to demand higher frequencies and lower power, PLLs with their hybrid and adaptive designs are set to drive the next wave of scalable, high-performance solutions for wireless and digital systems.

## 3. System View

This section dives into the world of the PLL, a crucial closed-loop control system. Its primary job is to precisely synchronize the frequency and phase of an output signal with a reference signal. We will start by breaking down the basic PLL framework, including its core components and their individual roles, along with how they work together. You will also learn about the distinct operational states a PLL can be in.

To better understand how PLLs behave in real-world scenarios, especially when dealing with unwanted interference, this paper presents a system model that includes the impact of noise, supported by relevant mathematical formulations.

Finally, this paper explores the various architectural flavors of PLLs, such as analog, digital, and all-digital PLLs. For each, this work discusses the trade-offs they present in terms of power efficiency, silicon area, locking speed, and scalability for advanced process technologies. This foundational understanding will pave the way for a more in-depth exploration of noise-aware PLL design and system-level optimization strategies.

### 3.1. Basic Framework

Understanding the Core of a Phase-Locked Loop: A Phase-Locked Loop (PLL) is a powerful feedback system designed to precisely synchronize the phase and frequency of an output signal with a reference signal [2,3]. This makes it a cornerstone in many applications, from frequency synthesis and signal modulation to wireless communications. As illustrated in Figure 1, a basic PLL is made up of five key components:Phase/Frequency Detector (PFD): This component acts as the “comparator”. It takes the input reference signal (
$f_{REF}$
) and compares its phase and frequency with the feedback signal from the Voltage-Controlled Oscillator (VCO). The PFD then generates an error signal that is directly proportional to any difference between the two.Charge Pump (CP): The Charge Pump converts this error signal into a proportional current or voltage.Loop Filter (LF): This is typically a low-pass filter that takes the charge pump’s output. Its job is to smooth out the signal by suppressing high-frequency noise, producing a stable control voltage (
$V_{ctrl}$
).Voltage-Controlled Oscillator (VCO): The VCO is the heart of the PLL’s output. It generates the output signal (
$f_{OUT}$
), and its frequency is directly controlled by the stable voltage from the Loop Filter. The relationship is often expressed as 
$f_{OUT}=f_{0}+K_{VCO}V_{ctrl}$
, where 
$f_{0}$
 represents the VCO’s free-running frequency and 
$K_{VCO}$
 is its gain.Frequency Divider (FD): In the feedback path, the Frequency Divider scales the VCO’s output frequency down to match the reference frequency. This allows for both integer-N and fractional-N configurations, providing flexibility in frequency generation.Together, these components create a sophisticated feedback loop that continuously adjusts the VCO’s output until it is perfectly locked in phase and frequency with the input reference. Through this closed-loop operation, a PLL dynamically moves through three distinct states [2]:

Free-Running State: When there is no reference signal, the VCO operates at its inherent free-running frequency (
$f_{0}$
). The PLL remains unlocked, and its output is independent of any external input.Capture State: As soon as a reference signal is introduced, the PLL enters the Capture State. Here, the PFD generates an error signal that begins to nudge the VCO’s frequency toward the reference frequency. The Loop Filter is crucial during this phase, smoothing out the control voltage (
$V_{ctrl}$
) and steadily reducing the frequency and phase differences until the VCO’s output gets close to the reference.Locked State: Once the phase error is sufficiently minimized (approaching zero, or 
$\Phi_{error}\approx 0$
), the PLL achieves the Locked State. In this state, the output signal is fully synchronized in both phase and frequency with the reference. The PLL maintains this lock by continuously making subtle adjustments to the VCO, responding to even tiny variations in the reference signal to ensure robust stability.
It is important to note that significant disturbances, like sudden changes in the reference frequency or a complete loss of the reference signal, can disrupt this synchronization. Such events may cause the PLL to fall back into the capture or even the free-running state.

This dynamic interplay of feedback and state transitions highlights the PLL’s remarkable ability to provide precise synchronization, making it an indispensable component in applications like frequency synthesis, signal modulation, and wireless communications.

### 3.2. System Model

The transient response of a PLL is generally a nonlinear process that cannot be formulated easily. But once phase locking is achieved, a linear approximation can be used to gain intuition [2,4]. Figure 2 shows a linear phase-domain model for the classical PLL, where K_d_ is the PFD detection gain. F_LFP_(s) is the loop filter trans-impedance transfer function, and K_VCO_ is the VCO tuning gain in rads/V and frequency divider ratio (*N*). Additionally, various noise sources include reference noise (
$\Phi_{REF,n}$
), divider noise (
$\Phi_{DIV,n}$
), PFD noise (
$\Phi_{PFD,n}$
), charge pump noise (
$i_{CP,n}$
) and VCO noise (
$\Phi_{VCO,n}$
). The open-loop transfer function can be calculated as Equation (Equation 1).


**Understanding PLL Behavior–Linear Approximation and Noise:**


The initial transient response of a PLL is complex and inherently nonlinear, making it challenging to model precisely. However, once the PLL achieves phase lock, we can simplify its behavior using a linear approximation. This provides a valuable intuitive understanding of how the system operates. Figure 2 illustrates a linear phase-domain model for a classical PLL. In this model:K_d_ represents the PFD (Phase/Frequency Detector) detection gain.F_LFP_(s) is the loop filter’s trans-impedance transfer function.K_VCO_ is the VCO (Voltage-Controlled Oscillator) tuning gain in radians per volt (rad/V).*N* is the frequency divider ratio.This model also accounts for various noise sources that can impact PLL performance:
$\Phi_{REF,n}$
: Reference noise;
$\Phi_{DIV,n}$
: Divider noise;
$\Phi_{PFD,n}$
: PFD noise;
$i_{CP,n}$
: Charge pump noise;
$\Phi_{VCO,n}$
: VCO noise.The open-loop transfer function of this linearized PLL can be calculated using Equation (Equation 1). This analysis sets the stage for understanding how different components and noise sources affect the PLL’s overall performance.
(1)
$$G_{(}s)=\frac{K_{d}.F_{LPF}\left(s\right).K_{VCO}}{s.N}$$
With Equation (Equation 2) is the noise transfer function for VCO noise to PLL output: 
(2)
$$H_{VCO}\left(s\right)=\frac{1}{1+G(s)}$$
This indicates that VCO noise is high-pass filtered, suppressing low-frequency components. The loop phase noise, encompassing contributions from the reference, divider, PFD, and charge pump, is referred to the divider input as Equation (Equation 3).
(3)
$$P_{loop}\approx \frac{S_{\Phi loop,n}}{2}=\frac{1}{2}.N^{2}.\left(S_{\Phi REF,n}+S_{\Phi DIV,n}+S_{\Phi PFD,n}+\frac{S_{iCP,n}}{K_{d}^{2}}\right)$$

where 
$S_{\Phi \;loop,n}$
 denotes the phase noise power spectral density, and 
$P_{loop}$
 is the single-sideband noise power-to-carrier ratio. The noise transfer function for loop phase noise is calculated by Equation (Equation 4).
(4)
$$H_{loop}\left(s\right)=\frac{G(s)}{1+G(s)}=1-H_{VCO}\left(s\right)$$
This shows that loop phase noise is low-pass filtered. By Equation (Equation 5), the total output phase noise combines these contributions: 
(5)
$$\Phi_{\mathrm{out}}\left(s\right)=H_{\mathrm{loop}}\left(s\right)\cdot \left(\Phi_{\mathrm{ref},n}\left(s\right)+\Phi_{\mathrm{div},n}\left(s\right)+\Phi_{\mathrm{PFD},n}\left(s\right)+\frac{i_{\mathrm{CP},n}\left(s\right)}{K_{d}}\right)+H_{\mathrm{VCO}}\left(s\right)\cdot \Phi_{\mathrm{VCO},n}\left(s\right)$$
Both transfer functions share the same 3-dB bandwidth, defined by 
$|G(j2\pi f_{c})|=1$
, approximating to Equation (Equation 6): 
(6)
$$f_{c}\approx K_{d}\cdot K_{\mathrm{VCO}}\cdot F_{\mathrm{LPF}}\left(0\right)$$
For a first-order loop filter, this model enables optimization of loop parameters to balance noise suppression, bandwidth, and stability, critical for PLL design in practical applications.

### 3.3. Classification of PLL Architecture

Phase-Locked Loops (PLLs) are indispensable in modern communication and signal processing, with various classifications tailored to specific design principles and application needs [1,2,3,4]. Broadly, PLLs fall into two main categories:Analog PLLs (APLLs): These utilize continuous-time components, offering high precision and often lower noise in certain applications.Digital PLLs (DPLLs): Employing discrete-time processing, DPLLs excel in superior integration, scalability, and programmability, especially in advanced semiconductor processes.
Within these categories, the choice of frequency division method significantly impacts performance:

Integer-N PLLs: These divide the VCO frequency by an integer, offering straightforward design but with limited frequency resolution.Fractional-N PLLs: These use advanced techniques like sigma-delta modulation to achieve much finer frequency resolution, though they can introduce more complex noise characteristics.
Specialized PLL designs cater to a wide range of demands:

Charge-Pump PLLs (CP-PLLs): A very common type, known for their robust locking behavior and relatively simple implementation.All-Digital PLLs (ADPLLs): These push the boundaries of digital integration, aiming for complete digital control to maximize scalability and minimize analog component sensitivities.Sub-Sampling PLLs (SS-PLLs): Designed for low power consumption by directly sampling the reference clock, which can reduce noise contributions.
The loop filter configuration (passive or active) further refines system stability and bandwidth, which are critical for robust operation. Beyond these fundamental types, advanced variants address specific high-performance requirements:

Injection-Locked PLLs (ILPLLs): These leverage injection-locking mechanisms for superior low-jitter performance and rapid locking times, often used in high-speed communication.Delay-Locked Loops (DLLs): While similar in principle, DLLs typically synchronize phase and delay rather than frequency, commonly used for clocking applications in digital systems to reduce clock skew.
The selection of a specific PLL architecture is a crucial decision, heavily dependent on key considerations such as phase noise performance, power efficiency, frequency agility, and integration feasibility. This remarkable versatility solidifies PLLs as an indispensable cornerstone of modern electronics, continuously driving innovation across various industries.

#### 3.3.1. Analog Phase Locked Loops (APLLs)


**Precision and Performance**


Analog Phase-Locked Loops (APLLs) are indispensable for high-precision frequency synthesis and synchronization, delivering continuous time phase alignment critical for applications demanding ultra-low phase noise and spectral purity, such as 5G RF transceivers, radar systems, analog demodulation (e.g., FM/AM radio), and clock synchronization in high-speed microprocessors [2,4], as shown in Figure 3.


**Core Components and Operation**


An APLL integrates several key components to achieve its precision:Phase Detector (PD): Typically a Gilbert cell or double-balanced mixer, the PD compares the reference phase (
$\Phi_{IN}$
) with the VCO’s output phase (
$\Phi_{OUT}$
). It generates an error voltage proportional to their difference (
$\Delta \Phi =\Phi_{IN}-\Phi_{OUT}$
). These detectors often have a gain (
$K_{d}$
) of 0.1–1 V/rad and can achieve 10–20 ps resolution.Loop Filter (LF): This component processes the error voltage from the PD. It is often a second-order passive RC network (e.g., 10 pF/1 k
Ω
, with a 50–200 kHz bandwidth) or an active OTA-based filter designed for a 45–60-degree phase margin. The LF’s job is to suppress high-frequency noise, producing a stable control voltage (
$V_{ctrl}$
).Voltage-Controlled Oscillator (VCO): The VCO generates the PLL’s output signal. Its frequency (
$f_{OUT}$
) is controlled by 
$V_{ctrl}$
, following the relationship 
$f_{OUT}=f_{0}+K_{VCO}*V_{ctrl}$
, typically ranges from 50–200 MHz/V. Common VCO types include the following:–LC oscillators: These offer superior phase noise performance (e.g., −120 to −130 dBc/Hz at 1 MHz offset) and a 10–15% tuning range (1–10 GHz).Phase noise in LC oscillators, and oscillators in general, quantifies the spectral purity of the output signal. An ideal oscillator would produce a perfect single frequency, represented by an impulse in the frequency domain. However, real-world oscillators, including LC oscillators, are affected by various noise sources (e.g., thermal noise, flicker noise from active devices). These noise sources cause random fluctuations in the phase of the output signal, which manifest as “skirts” or sidebands around the main carrier frequency in the power spectrum. The unit of measurement for phase noise is typically dBc/Hz (decibels below the carrier per Hertz).∗dBc (decibels below the carrier): This part of the unit indicates the power of the noise relative to the power of the main carrier signal, expressed in decibels. A lower dBc value signifies better phase noise performance (less noise relative to the signal). For example, −90 dBc means the noise power is 90 dB lower than the carrier power.∗Hz (per Hertz): This part refers to the measurement bandwidth. Phase noise is measured within a specific 1 Hz bandwidth at a certain offset frequency from the carrier. This normalization to a 1 Hz bandwidth allows for a standardized comparison of phase noise performance across different devices and measurement setups.–Compact ring VCOs: While smaller (0.01–0.05 mm^2^) and more power-efficient (1–2 mW), they exhibit slightly higher phase noise (−100 to −110 dBc/Hz).An APLL operates through dynamic states:–Free-running: The VCO operates at its free-running frequency (
$f_{0}$
, e.g., 1–6 GHz) before lock.–Capture: The PLL actively steers 
$f_{OUT}$
 toward the reference frequency (within 10–100 MHz) based on the loop bandwidth and 
$K_{VCO}$
.–Phase-locked: The system achieves synchronization with sub-1-degree phase error and fast lock times of 10–50 microseconds.


**Stability, Noise, and Performance Analysis**


APLLs maintain synchronization well against minor reference variations (e.g., ±100 kHz). However, significant disturbances (e.g., abrupt changes >100 kHz) can cause them to lose lock and revert to the capture or free-running state, necessitating robust loop design [2,4].

Mathematically, the open-loop transfer function is G(s), from which the closed-loop transfer function is given by 
$H_{\mathrm{loop}}\left(s\right)=\frac{G(s)}{1+G(s)}$
. For a second-order RC filter, the loop filter transfer function is 
$F_{\mathrm{LF}}\left(s\right)=\frac{1+s\tau_{2}}{s(\tau_{1}+\tau_{2})}$
, where 
$\tau_{1}=R_{1}C_{1}$
 and 
$\tau_{2}=R_{2}C_{2}$
.

In terms of noise, VCO noise is high-pass filtered (
$H_{\mathrm{VCO}}\left(s\right)=\frac{1}{1+G(s)}$
), while loop noise (from the 
$S_{\Phi ,\mathrm{PD},n}$
, reference: 
$S_{\Phi ,\mathrm{REF},n}$
) is low-pass filtered by the closed-loop transfer function (
$H_{\mathrm{loop}}\left(s\right)=\frac{G(s)}{1+G(s)}$
). The output loop noise can be approximated as 
$P_{\mathrm{loop}}\approx \frac{1}{2}\left(S_{\Phi ,\mathrm{PD},n}+S_{\Phi ,\mathrm{REF},n}\right){\left|H_{\mathrm{loop}}\left(j2\pi f\right)\right|}^{2}$
. The loop bandwidth (
$f_{c}$
, typically 50–200 kHz) is critical for balancing noise suppression and lock time.


**Advantages and Challenges**


APLLs truly excel in RF applications due to their exceptional phase noise (<100 fs RMS jitter) and spur-free outputs. This makes them ideal for frequency synthesizers in 5G base stations, radar signal processing, and high-speed data links (e.g., 28 Gb/s SerDes).

However, APLLs face several challenges:PVT Sensitivity: Their performance can drift by 5–10% in 
$f_{c}$
 due to component variations from aging, temperature (−40 to 125 °C), or manufacturing tolerances.Noise Susceptibility: Thermal and flicker noise in the Loop Filter and VCO can add 1–3 dB to in-band noise.Poor Scalability: In sub-28-nm CMOS processes, APLLs typically require larger areas (0.1–1 mm^2^) and consume more power (5–20 mW) compared to their digital counterparts (0.01–0.05 mm^2^, 0.5–2 mW).Complex Analog Design: Fine-tuning parameters like 
$K_{d}$
, 
$K_{VCO}$
, and Loop Filter characteristics requires significant expertise.

To mitigate these disadvantages, engineers are exploring various strategies:PVT sensitivity can be reduced to less than 2% by integrating adaptive calibration circuits (e.g., varactor banks or automatic gain control).Noise susceptibility can be improved by 3–5 dB through kT/C noise cancellation or by 2–4 dB using high-Q LC tanks.Scalability can be enhanced by adopting hybrid PLL architectures that combine analog low-noise components with digital calibration for advanced process nodes.Design complexity can be alleviated with automated design tools that optimize loop parameters.

In conclusion, APLLs remain unmatched for high-frequency RF systems (1–10 GHz) due to their spectral purity. However, their limitations in modern System-on-Chips (SoCs) often favor digital or hybrid PLL approaches. This necessitates continued innovation in noise mitigation and integration techniques to ensure their relevance in emerging technologies like 6G and quantum computing applications.

#### 3.3.2. Digital Phase Locked Loops (DPLLs)

Precision and Flexibility through Digital Control: Digital Phase-Locked Loops (DPLLs) represent a significant evolution from their analog counterparts [11,14,23]. They replace traditional analog components with digital circuitry, leveraging the power of digital signal processing (DSP) to boost precision and flexibility, as seen in Figure 4.

Instead of a conventional analog phase detector, DPLLs employ digital phase detectors that discretely sample and compare the phase of the input and feedback signals. The resulting error signal is then processed by a digital filter. This digital approach allows for precise mathematical manipulation of control parameters, effectively eliminating issues commonly associated with analog component tolerances and environmental variations. The filtered control signal then adjusts a numerically controlled oscillator (NCO), which digitally generates the output frequency.


**Key Advantages of DPLL Architecture**


This digital architecture offers several compelling advantages:Higher immunity to noise: Digital signals are generally more resilient to electrical interference.Improved stability: DPLLs exhibit enhanced stability across temperature changes and over time (aging), unlike analog designs.Software reconfigurability: Their digital nature allows them to be easily reconfigured or tuned through software, providing immense flexibility.

DPLLs are now widely adopted in modern digital communication systems, software-defined radios, and frequency synthesis applications. Their adaptability and seamless integration with other digital processing units are crucial in these areas. Unlike APLLs, which rely on continuous analog feedback, DPLLs operate in a discrete-time manner, making them less susceptible to the variability of analog components and external environmental factors.


**Trade-offs and Considerations**


However, the discrete-time nature of DPLLs introduces some trade-offs:Processing delay: While digital control enhances precision and stability, it can introduce a processing delay, potentially limiting the response speed compared to APLLs.Quantization noise: The reliance on digital computations can lead to quantization noise, necessitating sophisticated filtering techniques to maintain signal integrity.Design complexity and power consumption: The design of DPLLs often involves complex digital algorithms for phase detection and loop filtering, which require computational resources and can increase power consumption.


**Programmability and Integration**


One of the standout advantages of DPLLs is their programmability. Since their loop parameters are controlled by digital algorithms, they can be easily adjusted to accommodate different frequency ranges and application requirements without hardware modifications. This inherent flexibility makes DPLLs exceptionally well-suited for applications demanding dynamic frequency synthesis, adaptive clock recovery, and phase synchronization in digital communication networks.

Furthermore, integrating DPLLs into modern semiconductor processes is more straightforward. Digital circuits scale efficiently with advancements in fabrication technology, leading to lower power consumption and improved performance in compact System-on-Chip (SoC) designs.

Figure 5 illustrates the phase detection concept in a DPLL. In conventional analog PLLs (APLLs), the phase frequency detector (PFD) generates UP and DN pulses to drive a charge pump and loop filter, which in turn control the voltage-controlled oscillator (VCO). This analog-based approach offers low phase noise and stable performance, and is often adopted in hybrid APLL architectures that combine digital logic with analog components to extend the locking range and maintain noise performance. In fully digital PLLs (DPLLs); however, the conventional PFD is typically substituted by a time-to-digital converter (TDC), which directly quantizes the phase difference between the reference (
$V_{REF}$
) and feedback (
$V_{FB}$
) signals into a digital word. This eliminates the analog charge pump, making the loop fully digital while preserving the core function of precise phase and frequency comparison. The timing diagram in Figure 6 highlights how the digital phase detection in a DPLL, whether implemented as a classic PFD or a TDC, plays a central role in maintaining loop stability and performance [6].

#### 3.3.3. All Digital Phase Locked Loops (ADPLLs)—The Next Evolution

Building on the principles of DPLLs, All-Digital Phase-Locked Loops (ADPLLs) represent an even more advanced approach. These systems completely remove analog components, replacing them with fully digital elements [1,7]. This shift brings significant advantages in precision and programmability. The core components of an ADPLL, as shown in Figure 7, include the following:A Digital Loop Filter (DLF): The DLF processes phase error signals using arithmetic operations. This digital implementation allows for highly precise and programmable filtering characteristics, greatly enhancing stability and noise immunity by eliminating variations introduced by analog components. Figure 8 illustrates a typical model of an ADPLL’s digital loop filter.A Time-to-Digital Converter (TDC): This component measures the time difference (and thus phase difference) between the reference and feedback signals, converting it into a digital value.A Digitally Controlled Oscillator (DCO): Unlike a VCO, the DCO’s frequency is directly controlled by digital codes from the DLF, generating the output signal entirely in the digital domain.

This all-digital architecture pushes the boundaries of integration and performance, offering benefits like increased scalability with advanced semiconductor processes and reduced sensitivity to environmental factors.

The Time-to-Digital Converter (TDC) [8] is a critical component in all-digital PLLs (ADPLLs), responsible for precisely measuring the phase differences between the input and feedback signals. TDCs convert these analog time differences into digital values that the rest of the ADPLL can process. There are two primary techniques for implementing TDCs:Linear TDCs: These provide high-resolution phase measurements, offering superior precision in detecting minute phase differences. However, this precision often comes at the cost of increased complexity in their implementation. An example of a linear TDC is shown in Figure 9.Bang-Bang TDCs: In contrast, bang-bang TDCs employ a binary decision approach to adjust the oscillator. They essentially determine if the feedback signal is leading or lagging the reference, providing a “bang” (too fast) or “bang” (too slow) signal. This method offers a simpler and more power-efficient solution, making them suitable for many practical applications where extreme precision is not strictly necessary. Figure 10 illustrates a bang-bang TDC [24].

While linear TDCs excel in their ability to resolve fine phase distinctions, bang-bang TDCs offer a compelling trade-off, balancing performance with reduced complexity and power consumption.

The digitally controlled oscillator (DCO) is a cornerstone of all-digital PLLs (ADPLLs), completely replacing traditional voltage-controlled oscillators (VCOs) by generating its frequency output entirely through digital means, as illustrated in Figure 11. DCOs can be designed in a couple of ways:Explicit Digital-to-Analog Converters (DACs) driving an analog VCO: In this approach, a DAC converts the digital control word into an analog voltage, which then tunes a conventional analog VCO.Embedded DAC approach: This method directly integrates the digital tuning mechanism within the oscillator itself. Here, digital codes directly control the oscillator’s parameters, eliminating the need for a separate, explicit DAC.The embedded DAC method often proves to be more advantageous for low-power applications, such as those found in mobile and wireless communication. It typically leads to lower power consumption and better integration with digital logic, making it an ideal choice for compact and energy-efficient designs.

ADPLLs offer significant advantages over traditional, analog-heavy PLL architectures, making them increasingly popular in modern electronic design:Reduced Area: By eliminating bulky analog components, ADPLLs occupy a smaller silicon area. This is crucial for highly integrated System-on-Chip (SoC) designs where space is at a premium.Enhanced Power Efficiency: A key benefit of ADPLLs is their immunity to leakage currents, which are common in analog circuits. This directly translates to more power-efficient operation, a critical factor for battery-powered devices and energy-conscious systems.Robustness against PVT Variations: ADPLLs are remarkably robust against Process, Voltage, and Temperature (PVT) variations. Unlike analog designs that can be sensitive to manufacturing tolerances, voltage fluctuations, or temperature changes, ADPLLs ensure consistent and reliable performance across diverse operating conditions.Simplified Design Process: The design flow for ADPLLs is streamlined. The digital loop filter (DLF) can be directly constructed by transforming an s-domain filter (used in analog control theory) into its z-domain equivalent (for digital systems). This simplification allows for easier and more efficient design.Higher-Order Filtering: The digital nature of the DLF also allows for the ready implementation of higher-order filtering. This provides greater flexibility in shaping the loop’s response, enabling better noise suppression and improved stability.Adaptive Coefficient Tuning: ADPLLs excel in their ability to perform adaptive coefficient tuning. This means their operational parameters can be preset at power-up for optimal initial performance. More importantly, they can undergo real-time adjustments during operation to dynamically achieve either fast locking (for quick synchronization) or ultra-low jitter (for high-precision timing), depending on the application’s immediate requirements.

These combined advantages solidify ADPLLs as a superior choice for many contemporary and future electronic systems, offering a powerful blend of small size, efficiency, robustness, and flexibility.


**Challenges and Trade-offs of ADPLLs**


Despite their numerous advantages, all-digital PLLs (ADPLLs) are not without their limitations. Understanding these challenges is crucial for selecting the appropriate PLL architecture for a given application:Quantization Noise: The inherent reliance on digital processing introduces quantization noise. This can degrade the overall phase noise performance if not carefully managed through sophisticated design techniques and filtering.TDC Resolution and Power Consumption: The accuracy of Time-to-Digital Converters (TDCs) directly impacts the overall precision of the ADPLL. Achieving very high resolution in TDCs often requires complex, high-performance circuits, which can, in turn, increase power consumption.Finite Frequency Resolution: Due to the discrete nature of digital control, ADPLLs have a finite frequency resolution. This can limit their ability to achieve the ultra-fine tuning capabilities that are sometimes possible with analog counterparts.Accumulated Digital Jitter: ADPLLs may experience increased jitter due to the accumulation of digital errors over time. Careful design, including robust clocking and error correction mechanisms, is necessary to mitigate this.

Table 1 provides a comprehensive comparison between charge pump PLLs (CP-PLLs) and all-digital PLLs (ADPLLs), highlighting their respective strengths and weaknesses across various performance metrics. This comparison can guide designers in making informed choices based on their specific system requirements.

#### 3.3.4. Integer-N PLL


**Design, Performance, and Limitations**


Integer-N PLLs (as seen in Figure 12) are a type of frequency synthesizer where the output frequency is always an integer multiple of the reference frequency [6,18]. This design approach offers simplicity in both design and implementation, often leading to lower phase noise compared to other PLL architectures. Because the division factor in the feedback loop is consistently an integer, the loop remains stable and avoids complex fractional spurs. This makes integer-N PLLs highly suitable for applications that need low-noise and stable frequency synthesis, such as in wireless communication systems and high-frequency oscillators.


**Key Components and Operational Challenges**


To achieve optimal performance, integer-N PLLs require careful design of their key components: the voltage-controlled oscillator (VCO), dual-modulus divider, phase-frequency detector with charge pump (PFD/CP), and loop filter.

The dual-modulus divider typically uses a pulse-swallow technique. In this method, a prescaler initially divides the frequency by N+1 until a “swallow counter” reaches its programmed value. It then switches to dividing by N until the “program counter” completes its cycle. While effective for frequency synthesis, this mechanism results in a fixed output frequency step size that is equal to the reference frequency, which inherently limits the frequency resolution.

A major drawback of integer-N synthesizers is that the output frequency step is constrained by the reference frequency. This makes achieving fine frequency resolution difficult without reducing the reference frequency (
$f_{REF}$
), which, in turn, slows down the frequency locking process.

Moreover, narrow channel spacing often necessitates a low loop bandwidth. This reduces the PLL’s ability to suppress the VCO’s phase noise while simultaneously amplifying reference phase noise significantly, degrading spectral purity. This presents challenges in high-performance applications like RF and microwave systems. Integer-N synthesizers also lack flexibility in adapting to different reference crystal frequencies, limiting their applicability in multi-standard environments [6,18].


**Performance Enhancements and Inherent Limitations**


To address some of these issues, several performance enhancement techniques have been developed:Up/down skew reduction and up/down current mismatch reduction help improve charge pump linearity, which minimizes reference spurs and enhances phase noise performance.Using a sampling loop filter effectively suppresses charge pump noise, reducing the impact of reference phase noise on the system.

Despite these enhancements, integer-N PLLs have significant limitations in terms of frequency resolution. Since the output frequency is restricted to discrete multiples of the reference clock, achieving fine frequency steps would require an extremely high reference frequency, which is often impractical. Additionally, this constraint can lead to an increased phase noise floor due to the high-frequency division ratios needed for fine-resolution applications. These limitations make integer-N PLLs less suitable for applications demanding fine frequency resolution and frequency agility, such as modern wireless communication systems, where precise frequency tuning is essential.

#### 3.3.5. Fractional-N PLL—Achieving Finer Frequency Resolution

Fractional-N PLLs build upon the foundation of integer-N PLLs, but with a clever twist: they dynamically vary the frequency divider ratio between N and N+1. This innovative technique allows for non-integer division, which in turn enables finer frequency resolution without needing an extremely high reference clock frequency [13,14,15,25].

Unlike integer-N PLLs, which are limited to output frequencies that are discrete integer multiples of the reference, fractional-N PLLs offer a significantly wider range of frequency synthesis. This makes them indispensable in modern communication standards like GSM, LTE, and Wi-Fi, where precise and flexible frequency tuning is absolutely crucial.

Figure 13 illustrates the block diagram of a fractional-N PLL, showcasing how this dynamic division is integrated into the system.


**Challenges and Advantages of Fractional-N PLLs**


Fractional-N PLLs offer significant advantages in frequency synthesis, but they also come with a notable challenge: unwanted spurious signals, or spurs. These spurs are caused by the periodic toggling of the frequency divider ratio between N and N + 1. This toggling effectively modulates the VCO frequency, introducing phase noise and undesirable sidebands that can degrade the quality of the output signal.


**Mitigating Spurs with Delta-Sigma Modulation**


Combating Fractional Spurs with Delta-Sigma Modulation: In fractional-N Phase-Locked Loops (PLLs), a significant challenge arises from the generation of fractional spurs. These undesirable spurious tones appear in the output spectrum due to the non-integer division ratio, which inherently introduces a periodic pattern in the divider’s operation. If left unaddressed, these spurs can degrade the spectral purity of the output signal, leading to increased interference and reduced performance in communication systems. To effectively mitigate these problematic spurs, delta-sigma (
ΔΣ
) modulation is a crucial technique employed in fractional-N PLLs. This method operates by intelligently randomizing the toggling pattern of the divider. Instead of a fixed, repetitive division sequence, the 
ΔΣ
 modulator dynamically adjusts the instantaneous division ratio around the desired fractional value. For example, to achieve an average division of 3.5, the modulator might alternate between dividing by 3 and dividing by 4 in a carefully controlled, randomized sequence. By introducing this controlled randomness, 
ΔΣ
 modulation effectively spreads the quantization noise, which would otherwise manifest as distinct, coherent spurs over a much broader frequency spectrum. This process is akin to converting a few loud, piercing tones into a more diffuse, lower-level hiss. The discrete energy of the spurs is transformed into broadband noise, which is significantly easier to filter out using low-pass filters or other signal processing techniques.

Ultimately, the application of 
ΔΣ
 modulation in fractional-N PLLs yields substantial benefits: it significantly enhances the spectral purity of the output signal and minimizes interference in sensitive communication systems, allowing for more robust and reliable data transmission [13,14,15,25].


**Key Advantages**


Despite the challenge of spurs, fractional-N PLLs offer several compelling advantages that make them highly desirable for modern applications:Faster Frequency Settling Times: They can lock onto a new frequency much more quickly than integer-N PLLs.Increased Loop Bandwidth: A wider loop bandwidth allows for better tracking of reference variations and faster response.Superior Suppression of VCO Phase Noise: They are more effective at attenuating the intrinsic noise generated by the VCO.Reduced Amplification of Reference Phase Noise: They are less prone to amplifying noise originating from the reference clock.

These features make fractional-N PLLs particularly well-suited for modern wireless communication systems that demand both agility (the ability to quickly change frequencies) and stability (maintaining a clean and precise output).


**Design Complexity and Considerations**


However, the sophistication of fractional-N PLLs comes with increased design complexity. Implementing delta-sigma modulation requires advanced digital processing techniques, which can lead to higher power consumption and increased computational overhead. Furthermore, to maintain optimal phase noise performance, careful design of filtering algorithms and the use of adaptive calibration techniques are essential. These considerations highlight the trade-offs involved in leveraging the superior frequency resolution and performance of fractional-N PLLs.


**The Evolving Landscape of Fractional-N PLLs**


Despite the inherent challenges associated with spurious noise and design complexity, the relentless advancement of semiconductor technology continues to drive more efficient and robust implementations of fractional-N PLLs. The integration of sophisticated digital correction algorithms and adaptive filtering mechanisms further elevates their performance, solidifying their position as the preferred choice in high-performance RF and mixed-signal applications.

As the demand for high-speed, low-power communication systems intensifies, fractional-N PLLs are poised for continued evolution. Their ongoing development will ensure their vital relevance in the next generation of wireless and embedded systems, where precise and agile frequency synthesis is paramount.


**Comparative Analysis: Fractional-N vs. Integer-N PLLs**


Table 2 provides a comparative overview of fractional-N and integer-N PLLs, highlighting their key characteristics and suitability for different applications.

The fractional-N PLL excels in its flexibility and high frequency resolution, enabling precise and continuous frequency adjustments. This makes it an ideal choice for modern, demanding applications such as telecommunications and 5G networks, where agile frequency management is absolutely paramount. However, its complex design and inherent susceptibility to spurious noise (a byproduct of its fractional division capabilities) necessitate the use of advanced mitigation techniques to optimize its performance.

In contrast, the integer-N PLL offers simplicity and superior stability, often exhibiting lower phase noise. These attributes make it well-suited for more basic applications like system clocks or fixed-frequency generation. However, its primary limitation lies in its coarser frequency resolution, making it less adaptable for scenarios requiring fine-grained frequency control.

#### 3.3.6. Injected Locked PLLs (IL PLLs)—Enhancing Performance Through Synchronization

Injection-Locked PLLs (IL-PLLs) [12] are vital components in modern communication and signal processing systems. They stand out due to their unique ability to significantly improve phase noise performance, synchronization accuracy, and power efficiency.

These specialized PLLs are widely deployed in high-performance applications such as wireless communication, radar systems, clock recovery circuits, and optical communication. By leveraging the fundamental principle of injection locking, IL-PLLs can achieve remarkably precise phase and frequency synchronization while simultaneously maintaining low jitter and excellent phase noise characteristics. Figure 14 illustrates the block diagram of a typical IL-PLL structure.


**Inside Injection-Locked PLLs: Components and Principles**


An Injection-Locked PLL (IL-PLL) relies on several key components working together to ensure stable and precise operation [12].


**Core Components**


Voltage-Controlled Oscillator (VCO): This is the heart of the IL-PLL, acting as the primary signal generator. Its frequency can be dynamically adjusted to meet synchronization needs.Injection Source: This provides an external reference signal that subtly influences the VCO’s operating frequency. This “injection” is central to the IL-PLL’s unique locking mechanism.Phase Detector (PD): The PD compares the phase difference between the feedback signal (from the VCO’s output) and the reference signal. It generates an error signal that guides the subsequent adjustments within the loop.Loop Filter: This component processes the error signal from the PD. It acts as a low-pass filter, removing high-frequency noise and ensuring a smooth, stable control voltage for tuning the oscillator.Optional Frequency Divider: In some IL-PLL configurations, a frequency divider is used. This component scales down the VCO’s frequency before it is injected back into the loop. This can improve stability and phase alignment, particularly for certain frequency ranges.

The feedback loop in an IL-PLL ensures continuous synchronization by dynamically adjusting the VCO’s output frequency in response to the injected reference signal.


**The Principle of Injection Locking**


The fundamental working principle of IL-PLLs centers on injection locking. This is a fascinating phenomenon where an oscillator’s natural (or “free-running”) frequency is pulled toward and eventually locked onto the frequency of an injected external reference signal [12].

This process occurs due to nonlinear coupling effects within the oscillator circuit. When an external signal is injected, it modifies the oscillator’s intrinsic phase dynamics. This creates a self-stabilizing effect that effectively forces the oscillator’s output to align itself with the injected reference.

Mathematically, this crucial behavior can be described by Adler’s equation (Equation (Equation 7)), which precisely expresses the phase evolution of the oscillator in response to the injection-locking phenomenon.
(7)
$$\frac{d_{\phi }}{dt}=\Delta \omega -\kappa sin\left(\phi \right)$$

where 
ϕ
 is the phase difference between the injected signal and the VCO’s natural oscillation, 
Δ

ω
 is the frequency offset, and 
κ
 represents the injection strength. When steady-state locking is achieved, the phase difference stabilizes, ensuring consistent frequency tracking and reduced phase noise.


**Key Advantages**


IL-PLLs offer several notable advantages over conventional PLL architectures:Improved Phase Noise Performance: By leveraging a high-quality reference signal through injection locking, IL-PLLs can significantly suppress phase noise contributions from the oscillator, leading to cleaner output signals.Lower Power Consumption: IL-PLLs can operate with reduced loop bandwidths, allowing for energy-efficient circuit designs and contributing to lower overall power consumption.Fast Locking Times: These systems are particularly useful in applications requiring rapid frequency acquisition, such as adaptive frequency synthesis and agile radar systems, due to their ability to achieve fast locking times.Reduced Jitter: A key advantage is the reduced jitter, as the injected reference signal dominates the oscillator’s phase response, resulting in enhanced timing stability.


**Limitations and Challenges**


Despite these advantages, IL-PLLs also have certain limitations that require careful management:Limited Locking Range: The frequency pulling effect depends on the injection strength and oscillator characteristics, meaning that IL-PLLs can only operate reliably within a narrow range of reference frequencies.Sensitivity to Injection Strength: Instability can arise if the injected signal is too weak, leading to unreliable phase locking.VCO Design Constraints: Oscillators must be specifically designed to exhibit strong injection-pulling characteristics while simultaneously maintaining low inherent noise, posing a significant design challenge.Distortion Effects: In high-noise environments, distortion effects may emerge due to harmonics and spurious tones introduced during the injection process.Dependence on External Reference: IL-PLLs rely on a high-quality external reference signal, making them dependent on external sources, which may not always be readily available or stable.Temperature Sensitivity: Temperature variations can affect locking performance, often necessitating additional compensation techniques to maintain stability across different operating conditions.


**Diverse Applications**


The applications of IL-PLLs span multiple industries where precision frequency control and low noise are essential:Wireless Communication: IL-PLLs are used for frequency synthesis in RF transceivers, providing the high spectral purity needed for modulation and demodulation processes.Clock Recovery Circuits: In high-speed digital systems, IL-PLLs help regenerate stable clock signals from noisy data streams, effectively reducing timing jitter.Radar Systems: They enable coherent signal generation, improving target detection accuracy through phase-sensitive processing.Optical Communication: IL-PLLs facilitate carrier recovery and phase synchronization in coherent optical receivers, enhancing data integrity and transmission efficiency.

The unique capabilities of IL-PLLs make them an invaluable tool for modern high-performance electronic systems.

#### 3.3.7. Delay Locked Loops (DLLs)—Precision Timing Circuits

A Delay-Locked Loop (DLL) [16], as shown in Figure 15, is a crucial timing circuit primarily used for phase synchronization rather than frequency synthesis. This sets it apart from a Phase-Locked Loop (PLL), which actively tunes an oscillator’s frequency to track a reference signal. Instead, a DLL precisely regulates the delay of a signal to align its phase with a reference clock.

This unique characteristic makes DLLs exceptionally beneficial in applications that demand minimal jitter, stable timing control, and accurate phase alignment. They are often found in high-speed digital systems where precise clocking is paramount.


**Understanding Delay-Locked Loops (DLLs): Structure, Advantages, and Limitations**


A Delay-Locked Loop (DLL) is a sophisticated timing circuit built around a feedback mechanism to achieve precise phase synchronization.


**How a DLL Works**


The structure of a DLL involves several key components working in concert [16,26]:Phase Detector (PD): This component measures the phase difference between the reference clock and the delayed output signal. It then generates an error signal indicating the necessary adjustment.Charge Pump (CP) and Loop Filter (LF): These work together to convert the phase error signal into a control voltage. The Loop Filter smooths this voltage, eliminating high-frequency noise, before applying it to the VCDL.Voltage-Controlled Delay Line (VCDL): The VCDL is where the magic happens. It introduces a controllable delay to the input signal, ensuring that the output phase gradually aligns with the reference clock.

Through this continuous feedback, the DLL dynamically corrects any phase errors until precise synchronization is achieved. Unlike PLLs, which dynamically alter frequency, DLLs operate by simply delaying an input clock signal until its phase matches the reference. This fixed-frequency operation enhances stability and prevents long-term phase noise accumulation, making DLLs particularly suitable for clock distribution and high-speed data communication.


**Advantages of DLLs**


Compared to PLLs, DLLs offer several compelling advantages:Superior Jitter Performance: By avoiding frequency drift, DLLs typically achieve better jitter performance.Faster Locking Time: Since they do not need to converge on a frequency, DLLs generally lock much faster than PLLs.Lower Power Consumption: DLLs do not require a Voltage-Controlled Oscillator (VCO), which is often a power-hungry component in PLLs, leading to lower overall power consumption.Improved Clock Distribution: They excel at minimizing clock skew and enhancing overall system reliability in integrated circuits.


**Limitations of DLLs**


However, DLLs also come with certain limitations:No Frequency Multiplication: DLLs cannot perform frequency synthesis or multiplication, making them unsuitable for applications that require generating new frequencies.Limited Operating Range: Their operating range is constrained by the VCDL’s delay adjustment window.Phase Ambiguity: Some DLL designs may exhibit phase ambiguity, necessitating additional logic to ensure reliable locking.


**Applications and Future Trends**


Despite these limitations, DLLs are widely used in applications demanding precise timing control, such as clock recovery in high-speed communications, DDR memory interfaces, and microprocessor synchronization. In some cases, hybrid PLL-DLL architectures combine the strengths of both circuits, optimizing both phase accuracy and frequency flexibility.

Overall, while PLLs excel in frequency synthesis, DLLs are invaluable in scenarios where stable phase alignment, low jitter, and power efficiency are paramount. This makes them essential components in modern computing and communication systems.

Table 3 provides a comprehensive comparative analysis of various Phase-Locked Loop (PLL) and Delay-Locked Loop (DLL) architectures. This includes Analog PLLs (APLLs), digital PLLs (DPLLs), all-digital PLLs (ADPLLs), integer-N PLLs, fractional-N PLLs, Injection-Locked PLLs (IL-PLLs), and Delay-Locked Loops (DLLs).

This evaluation highlights key differences across critical criteria such as Control Type, Phase Detector, Oscillator Type, Loop Filter, Phase Noise/Jitter, Lock Time, Design Complexity, Flexibility, and Typical Applications.

The data presented in the table reveals distinct trade-offs and strengths among these architectures:Analog-based architectures like the APLL generally offer simplicity in their fundamental design but are often characterized by limited flexibility.Digital solutions such as the ADPLL and DPLL provide enhanced precision and adaptability, particularly due to their programmability, though this comes with an increase in design complexity.Fractional-N PLLs stand out for their ability to achieve agile frequency synthesis, allowing for fine frequency tuning. However, this often comes at the cost of higher jitter compared to integer-N designs.DLLs are particularly notable for their low noise performance in applications requiring high-speed data synchronization, excelling in precise phase alignment rather than frequency generation.

This comprehensive comparison serves as a valuable guide for designers, enabling informed decisions when selecting the most appropriate timing and frequency control solution for diverse electronic systems.

## 4. Challenges and Solutions

Phase-Locked Loops (PLLs) are truly indispensable in today’s electronic systems. They form the backbone for critical functions like frequency synthesis, clock generation, and synchronization across a vast array of applications, including 5G communications, high-speed data converters, and ultra-low-power IoT devices.

While architecturally diverse, encompassing analog PLLs (APLLs), digital PLLs (DPLLs), integer-N PLLs (IPLLs), fractional-N PLLs (FPLLs), and charge-pump PLLs (CPPLLs), among others, PLLs uniformly face significant challenges. These challenges stem from their inherent sensitivity to non-idealities and the complex trade-offs that arise from their mixed-signal nature.

As semiconductor processes shrink to sub-7 nm nodes and System-on-Chip (SoC) complexity continues to escalate, PLL design demands unprecedented levels of precision, robustness, and efficiency. This section will provide a rigorous, system-level analysis of the primary challenges in PLL design. We will dissect their physical and architectural origins, quantify their impact, and explore cutting-edge strategies for their mitigation.

### 4.1. Phase Noise and Jitter


**The Critical Impact of Phase Noise and Jitter in PLLs**


Phase noise and jitter are two fundamental imperfections that significantly degrade the performance of Phase-Locked Loops (PLLs). This is especially true in demanding applications like high-speed communication, data conversion, and clock distribution systems.

Phase noise is essentially the frequency-domain representation of short-term, random fluctuations in an oscillator’s phase. It appears as a spectral spreading around the carrier frequency and primarily originates from thermal noise, flicker noise, and device noise within the Voltage-Controlled Oscillator (VCO) [25,27]. However, other loop components, such as the charge pump [7] and the reference clock [28], also contribute to phase noise.

In the time domain, these random phase fluctuations are quantified as jitter. Jitter measures the uncertainty or variability in the exact timing of output signal transitions. It is a critical metric, continuously being refined in advanced designs like those by Chae et al. [21] and Park et al. [14].


**Far-Reaching Implications**


The implications of phase noise and jitter are widespread and severe:Digital Systems: Jitter erodes crucial timing margins, potentially leading to setup/hold violations that cause data errors. This was observed in high-jitter early designs [25].Data Converters (ADCs/DACs): In analog-to-digital or digital-to-analog converters, jitter introduces aperture uncertainty, directly compromising the converter’s resolution [14].RF Systems: In radio frequency systems, phase noise limits adjacent channel rejection and significantly increases bit error rates. This is a major concern addressed in wideband PLL designs [22].The PLL’s loop bandwidth plays a pivotal role in managing these non-idealities. A wider bandwidth can improve the tracking of reference noise, but it also allows more VCO noise to “leak” through to the output [29]. Conversely, a narrower bandwidth, often used in low-jitter designs [15], is effective at filtering out VCO noise, but this comes at the cost of prolonged acquisition time and potentially increased jitter induced by the reference clock.


**Mitigation Strategies**


Mitigating phase noise and jitter requires a comprehensive, multifaceted approach:VCO Optimization: Employing high-Q LC-VCOs is crucial for minimizing the intrinsic noise generated by the oscillator itself [27,30].Charge Pump and Loop Filter Design: Optimizing the design of the charge pump and loop filter helps reduce ripple and current mismatch, which are significant sources of noise [7,9].Power Integrity: Utilizing differential signaling and robust power isolation techniques helps mitigate disturbances introduced by the power supply.Advanced Architectures: In modern designs like sub-sampling PLLs [31] and digital PLLs [21], innovative design techniques are used to shape or directly reduce noise contributions. Examples include the work by Thaller et al. [31] and Chae et al. [21].Ultimately, effective management of phase noise and jitter demands meticulous co-optimization across the entire PLL loop. These effects are intricately linked to both the analog and digital domains, a principle underscored by the success of hybrid approaches [11] and state-of-the-art performance demonstrated by recent work [23].

### 4.2. Loop Stability and Bandwidth Trade-Off

Loop stability and bandwidth configuration are foundational aspects of Phase-Locked Loop (PLL) design, directly impacting both the dynamic response and noise performance of the system. The loop bandwidth of a PLL dictates its responsiveness to changes in the input reference or other system disturbances. A wider bandwidth generally leads to the following:Faster tracking: The PLL can quickly lock onto and follow variations in the input signal.Reduced reference spur filtering: It is less effective at filtering out unwanted spurious signals from the reference.Lower in-band phase noise: It can better suppress noise within the operating band, as seen in wideband designs [22,27].However, this comes with a trade-off. A wider bandwidth also allows more of the VCO’s intrinsic noise to permeate the output, and it heightens the risk of loop instability [29]. Conversely, a narrower bandwidth, often used in low-jitter designs [21], offers several benefits:Enhanced VCO phase noise suppression: It more effectively filters out noise generated by the Voltage-Controlled Oscillator.Improved noise shaping: It can better shape the noise spectrum for desired performance.The drawbacks of a narrower bandwidth include sacrificing settling speed and tracking capability, as evidenced by the longer calibration times in some designs [32].


**Ensuring Loop Stability**


Loop stability is intrinsically linked to phase margin, a critical indicator of how close the system is to oscillating uncontrollably. Issues like poor loop compensation, inadequate phase margin, or overly ambitious bandwidth targets can lead to undesirable effects such as the following:Frequency response peaking: The PLL’s response becomes exaggerated at certain frequencies.Excessive jitter amplification: Unwanted timing variations are magnified.Potential instability: The loop might fail to maintain lock.These issues are particularly pronounced in high-order or fractional-N PLLs, where complex nonlinear and discrete-time behaviors make control dynamics more challenging [14,17]. To manage this inherent trade-off between performance and stability, designers frequently opt for Type-II second- or third-order PLL architectures. These provide enhanced control over the dynamic response and ensure adequate phase margin through meticulous loop filter design [7,11].


**Advanced Mitigation Techniques**


Advanced techniques, such as adaptive bandwidth control, offer a balanced approach. These systems dynamically adjust the bandwidth based on operating conditions:Widening it during acquisition: This allows for rapid locking [17].Narrowing it in steady state: This minimizes jitter once locked [15].Ultimately, achieving an optimal equilibrium between bandwidth and stability demands a deep, system-level understanding of noise sources, application requirements, and loop dynamics. This process is often supported by analytical modeling and extensive simulation across various process, voltage, and temperature (PVT) variations to ensure robust performance.

### 4.3. Spurs and Reference Spurious Tones

Spurs, or spurious tones, are deterministic spectral impurities that show up as discrete frequency components, offset from the main carrier signal. They are typically caused by multiples of the reference frequency or as artifacts of modulation. These unwanted tones severely degrade the spectral purity of Phase-Locked Loops (PLLs), leading to several problems:Interference in frequency synthesizers [33]..Degraded Signal-to-Noise Ratio (SNR) in wireless transceivers.Increased Bit Error Rates (BER) in multi-channel systems [26].Their impact is especially critical in 5G and Wi-Fi applications, where spectral regrowth from spurs can violate stringent emission masks, a concern effectively addressed in robust designs [11,34].


**Origins of Spurs**


Spurs can originate from various sources within a PLL:Reference Spurs: These arise from periodic disturbances, including the following:–Charge pump current mismatches.–Phase detector nonlinearities.–Reference clock leakage through parasitic paths [35].Fractional Spurs (in Fractional-N PLLs): The quantization noise from sigma-delta modulators, if not properly shaped, can fold back into the signal band, a known challenge in early fractional-N implementations [17].Layout-Induced Coupling: As integration density increases, especially in sub-16 nm processes, digital switching noise can infiltrate sensitive analog nodes, exacerbating spur levels. This was observed in a 16 nm PLL by Thaller et al. [24,31]. These effects intensify with growing integration density, posing significant hurdles in mixed-signal System-on-Chips (SoCs).


**Strategies for Mitigation**


Mitigating spurs requires a multifaceted strategy spanning circuit design, architectural choices, and careful layout:Charge Pump Linearization: Employing adaptive current matching techniques can reduce reference spurs to impressively low levels (e.g., below −75 dB), as demonstrated in a 65 nm CPPLL [7].Low-Glitch Phase Detectors: Pairing low-glitch phase detectors with fully differential signal paths helps minimize deterministic ripple [36].Advanced Sigma-Delta Modulation: In fractional-N PLLs, using high-order (e.g., fourth-order) sigma-delta modulators with advanced noise shaping can lower fractional spurs significantly (e.g., to −85 dB) [37,38].Robust Layout Strategies: Techniques like deep N-well isolation, guard rings, and shielded routing are crucial for effectively mitigating substrate noise. These methods have been validated in designs, including a 16 nm PLL [31].Effectively managing spurs is crucial for ensuring the high performance and reliability of PLLs in demanding modern electronic systems.

### 4.4. Frequency Range and Programmability

Modern communication and mixed-signal systems demand Phase-Locked Loops (PLLs) that offer both a wide frequency range and high frequency resolution. This is particularly evident in diverse applications like 5G, IoT, and mmWave technologies [22,34], where PLLs must support multiple operating standards or modes. These stringent requirements highlight the need for highly programmable PLLs capable of fine-tuning output frequencies without significantly degrading phase noise performance [17,22].


**The Challenges of Wide Frequency Operation**


The main difficulty lies in maintaining optimal PLL performance across such an expansive frequency spectrum:VCO Nonlinearity and Noise: Voltage-Controlled Oscillators (VCOs) with broad tuning curves often exhibit 
$K_{VCO}$
 nonlinearity, reduced gain control, and degraded phase noise, especially at the band edges [25,27].Fractional-N Complexity: In fractional-N PLLs, achieving fine resolution while effectively mitigating spurious tones introduces increasing design complexity [14,39].Loop Filter and Divider Consistency: Programmable loop filters and dividers must ensure loop stability and consistent behavior across all frequency settings, a critical aspect in multi-mode systems [34].


**Mitigation Strategies**


To address these significant challenges, designers employ a range of sophisticated techniques:Multi-band or Switched-Capacitor VCOs: These designs maintain linearity and noise performance by dynamically switching between different tuning banks. This approach is demonstrated in various works [30,40].High-Order Sigma-Delta Modulators: In fractional-N PLLs, leveraging high-order sigma-delta modulators provides exceptional frequency resolution while effectively suppressing unwanted spurs [17,41].Digital Control of Loop Parameters: Digital control enables mode-dependent reconfiguration of loop parameters, optimizing locking dynamics for different operational modes [21].Calibration Tables and Adaptive Tuning: Many systems implement calibration tables or adaptive tuning algorithms to automatically compensate for frequency drift and environmental variations, ensuring robust performance across diverse conditions [32,42].The synergy between analog programmability and digital flexibility is now pivotal in creating highly configurable, low-jitter PLLs essential for modern System-on-Chips (SoCs). This integrated approach is crucial for meeting the demands of next-generation communication and mixed-signal systems.

### 4.5. Process, Voltage, and Temperature (PVT) Variations

One of the most persistent challenges in PLL design is guaranteeing reliable operation across wide process, voltage, and temperature (PVT) variations. As semiconductor manufacturing shrinks to smaller nodes, process spread becomes more pronounced. This causes significant variations in critical circuit parameters like transistor threshold voltage, gain, and mobility [14,25,28]. Furthermore, PLLs often operate in environments prone to supply fluctuations and significant ambient temperature changes, particularly in demanding automotive or industrial settings, as highlighted by designs for robust applications like 5G and IoT [34].


**Impact of PVT Variations**


These variations do not just affect one part of the PLL; they impact multiple components simultaneously:The VCO frequency might drift.Loop filter characteristics could shift.Bias currents may deviate from their intended values.All these factors can lead to degraded locking behavior, compromised jitter performance, or even a complete loss of lock [26,29,43]. Moreover, because PLLs incorporate both analog and digital subsystems, they are particularly sensitive to mismatched scaling behavior across different temperature and supply domains, a challenge frequently observed in hybrid architectures [11].


**Mitigation Strategies for PVT Sensitivity**


Mitigating PVT sensitivity requires a combination of robust circuit design and adaptive calibration techniques:Circuit-Level Techniques: Designers widely employ techniques such as temperature-compensated biasing, bandgap references, and supply-independent current sources [9,11]. These aim to make individual components inherently less sensitive to environmental shifts.System-Level Calibration: Auto-calibration routines, often run at startup or periodically, are crucial for measuring and correcting process-induced mismatches, particularly within the VCO and charge pump [14,32].Digital Adaptation: Increasingly, digital calibration and feedback loops are being embedded directly into PLLs. These allow for real-time adaptation, ensuring consistent performance under dynamic operating conditions [21,42].Robust PVT tolerance is not just about product reliability; it is also critical for achieving high yield in mass production, especially in advanced nodes like 14 nm and 22 nm [14]. Mastering PVT variations is fundamental to the successful deployment of PLLs in modern electronic systems.

### 4.6. Area and Integration Complexity

As System-on-Chip (SoC) complexity continues to grow, integrating high-performance Phase-Locked Loops (PLLs) presents a significant challenge in terms of both silicon area and design intricacy. While PLLs are traditionally seen as compact blocks, several factors can dramatically increase their footprint and impose strict routing and layout constraints. These include the incorporation of on-chip inductors for LC-VCOs [25], high-resolution Digital-to-Analog Converters (DACs) [14], and programmable digital logic, as observed in designs like Wu et al. [41].

This challenge is even greater in multi-channel or multi-standard systems that require several PLLs. Duplicating large passive structures or repetitive control logic, a common issue in multi-core VCO designs [27], leads to excessive silicon consumption, higher power usage, and longer design timelines. Furthermore, integrating analog-sensitive PLL circuits alongside large digital cores worsens problems like substrate noise coupling and power domain conflicts, a concern highlighted in hybrid architectures [11].


**Strategies for Minimizing Area and Complexity**


To tackle these integration challenges, modern designs are increasingly favoring digital PLL (DPLL) architectures. These reduce or eliminate analog components and scale efficiently with advances in CMOS technology [9,21]. Other effective strategies include the following:Inductor-less oscillator topologies: Although these may have higher phase noise, they are used in applications where area and integration are prioritized over noise performance, such as in IoT-focused designs [44].Resource-sharing techniques: These involve using a single VCO across multiple PLLs or channels, proving effective in multi-output clock generators [38].Careful layout strategies: Techniques like analog/digital partitioning, guard rings, and localized decoupling are crucial in advanced nodes like 14 nm [14] to maintain signal integrity and minimize interference between different circuit blocks.Ultimately, minimizing area and complexity without compromising performance requires strong architectural foresight and seamless collaboration among circuit, layout, and system-level designers. This principle is underscored by the success of robust PVT-tolerant designs [42] and wideband solutions [22].

### 4.7. Power Consumption

Power efficiency is a paramount concern in modern Phase-Locked Loop (PLL) design, especially for mobile, always-on, and battery-powered systems where conserving energy is critical. Despite their essential role in clock and frequency generation, PLLs can consume a significant portion of a system’s total power. This is particularly true when they incorporate high-speed Voltage-Controlled Oscillators (VCOs) [25] and intricate digital control logic, as seen in designs like Wu et al. [41]. Key contributors to this power consumption include the following:VCO biasing circuitry [7].High-frequency dividers [28].Digital calibration or control blocks [14].These components often operate continuously or at gigahertz-range frequencies, as exemplified by the 10 GHz FMCW PLL [29].


**The Performance-Power Trade-Off**


The challenge of power efficiency is intensified by the necessity to uphold phase noise and jitter performance. Achieving superior performance in these areas typically demands elevated bias currents and rapid switching speeds. This often creates a direct trade-off, evident in low-jitter designs [14,21]. Moreover, in systems using LC-VCOs for their low-noise operation, the static current required to maintain oscillator startup and amplitude regulation can be a substantial energy cost, as noted in multi-core VCO implementations [27].


**Strategies for Power Reduction**


To address these power concerns, a combination of architectural and circuit-level innovations is essential:Oscillator Choice: For ultra-low-power applications, ring oscillators can replace LC-VCOs. While they might sacrifice some phase noise performance, they offer significantly reduced current consumption, a strategy demonstrated in IoT-focused designs [44].Power Management Techniques: Techniques like power gating, dynamic biasing, and clock gating prove highly effective in minimizing both dynamic (active) and static (leakage) power draw, contributing to more energy-efficient architectures.DPLL Optimization: In digital PLLs (DPLLs), optimizing logic paths and adopting event-driven architectures can significantly curtail switching activity, thereby reducing power consumption [9].Strategic Partitioning and Logic Design: The strategic partitioning of analog and digital power domains, coupled with the application of subthreshold or near-threshold logic, enhances energy efficiency without compromising functionality. This strategy has been refined in advanced process nodes like 14 nm [14] and 22 nm.By carefully implementing these techniques, designers can significantly improve the power efficiency of PLLs, making them suitable for an even wider range of energy-constrained applications.

### 4.8. Testing and Verification Complexity

The mixed-signal nature of Phase-Locked Loops (PLLs), coupled with their sensitivity to non-idealities and often prolonged lock times, makes testing and verification incredibly complex. This significantly impacts design cycles and increases production costs [44]. Validating a PLL’s performance across wide process, voltage, and temperature (PVT) variations, different frequency ranges, and various operating modes is a highly resource-intensive task, as clearly seen in the challenges encountered with advanced multi-mode designs [34].


**The Complexities of PLL Testing**


Several factors contribute to the intricate nature of PLL testing:Nonlinear Dynamics: Phenomena like fractional-N spur generation demand specialized test setups. Achieving sub-100 femtosecond (fs) jitter accuracy, crucial for high-precision PLLs, often requires measurement times exceeding 1 millisecond (ms) [14].Limited Observability: In modern SoCs below 28 nanometers (nm), limited access to internal nodes makes on-chip testing significantly more difficult.Mixed-Signal Interactions: The interplay between analog and digital components necessitates high-fidelity simulation models to accurately predict behavior, a concern addressed in hybrid architectures [11].Stringent Standards: Ensuring compliance with demanding standards, such as those for 5G, requires exhaustive corner-case testing, as demonstrated by the rigorous verification protocols of some designs [41].


**Mitigation Strategies**


To ease these complex testing challenges, several effective strategies are being employed:Built-in Self-Test (BIST) Circuits: Integrating BIST circuits, including on-chip jitter and spur measurement capabilities, streamlines the validation process. For example, a 40 nm PLL design achieved a significantly reduced test time of 10 microseconds (
μ
s) using BIST, lowering overall verification overhead [45].Mixed-Signal Simulation Frameworks: Utilizing frameworks that integrate both analog SPICE and digital Verilog simulations enhances accuracy, providing a comprehensive modeling approach [9].Automated Test Pattern Generation (ATPG): For digital PLLs (DPLLs), ATPG helps reduce test complexity. A notable example in a 14 nm DPLL design optimized test coverage and efficiency [14].These strategies collectively address the multifaceted testing demands of modern PLLs, aiming to make the verification process more efficient and cost-effective.

## 5. Comparison of Published Designs—Trends and Insights in PLL Design: A Decade of Advancements (2016–2025)

This section provides an insightful comparison of PLL designs from academic publications spanning 2016 to May 2025. PLL design involves a fundamental trade-off between jitter performance and power consumption. Lower jitter typically requires higher power through increased bias currents and larger device sizes, making it challenging to compare designs objectively across different applications and constraints. To address this challenge, Figure of Merit (FOM) provides a unified metric that combines both jitter and power into a single normalized parameter. The most commonly used jitter–power FOM is defined as Equation (Equation 8):
(8)
$$FOM=10log_{10}\left[{\left(\frac{Jitter}{1s}\right)}^{2}\times \left(\frac{Power}{1mW}\right)\right]$$

where 
Jitter
 is measured in seconds (s), 
Power
 is measured in milliwatts (mW), FOM is expressed in decibels (dB). This definition assumes jitter is measured as RMS deviation from ideal timing and power is evaluated at the operating frequency, ensuring a dimensionless normalized value. Lower FOM values indicate better overall performance by reflecting reduced jitter and/or lower power consumption. The scatter plots in Figure 16 illustrate how the proposed design of reference over the years achieves a competitive FOM compared to state-of-the-art alternatives, highlighting the trade-offs between jitter and power and validating the effectiveness of our reference choices. Table 4 summarizes the key specifications and design strategies (including only those with silicon results), while Figure 17 presents scatter plots of critical specifications for better visualization and understanding. Several conclusions can be drawn from this analysis:Steady Performance Improvement: PLL designs have shown consistent performance enhancements over the past decade in Figure 16. Integrated jitter has improved by approximately 2.5× per decade, decreasing from 0.16 psrms [25] to 65 fsrms [21], largely due to advancements in digital-to-time converter (DTC) calibration and noise suppression techniques. The Figure-of-Merit for jitter (FoM jitter) has also improved significantly, by 8.5 dB per decade, reaching −272 dB in a 2025 design [21]. Concurrently, bandwidth has expanded by roughly 10x per decade, with early designs operating at 2.4 GHz [6] and recent works achieving 9.05–37.0 GHz [22]. This progression clearly indicates an industry focus on achieving ultra-low jitter and wideband tunability.Process Technology’s Influence: Process technology plays a crucial role in PLL performance. Advanced nodes (14 nm and 22 nm), as utilized in designs like Wu et al. [14] and Dartizio et al. [39], enable the smallest silicon areas (e.g., <0.05 mm^2^), benefiting from aggressive scaling and optimized layouts. In contrast, 28–65 nm planar CMOS processes, employed in designs such as Huang et al. [26] and Jia et al. [46], tend to yield the highest FoM values (up to 196.9 dBc/Hz). This is attributed to the maturity of analog components and better power efficiency in these nodes. This suggests a clear trade-off: advanced nodes prioritize miniaturization, while mid-range nodes excel in noise performance.Architectural Diversity and Trade-offs: The PLL landscape encompasses a variety of architectures: Analog PLLs (APLLs), digital PLLs (DPLLs), all-digital PLLs (ADPLLs), integer-N, fractional-N, and Injection-Locked PLLs (IL PLLs). Fractional-N designs are prevalent, exemplified by Gao et al. [17] and Park et al. [14], offering superior frequency resolution (2.6–4.1 GHz) but often facing challenges with higher fractional spurs (−59 dBc). Integer-N PLLs, such as Kong et al. [6], provide good stability with lower jitter but are limited in tunability. It is evident that no single architecture consistently outperforms others across all metrics. However, ADPLLs show considerable promise in achieving high-frequency robustness, warranting further research and development.Multi-core VCOs and Harmonic-Shaping as Performance Benchmarks: Multi-core VCOs and harmonic-shaping techniques have emerged as key enablers for high-performance PLLs. Quad-core designs, such as those by Jia et al. [46] and Guan et al. [32], achieve impressive FoM values up to 200.2 dBc/Hz and bandwidths exceeding 35 GHz, while minimizing area (e.g., 0.049 mm^2^ in Gong et al. [47]). Techniques like harmonic extraction and mode-switching, as implemented in Guo et al. [40], further enhance phase noise suppression, making these approaches essential for wideband applications.DTC-based Techniques for Jitter and Spur Mitigation: DTC-based techniques are crucial for reducing jitter and spurs. Earlier implementations improved linearity through background calibration [37], while recent designs leverage quantization-error compensation to achieve remarkably low jitter, such as 65 fsrms [21]. However, the effectiveness of DTCs appears to diminish at bandwidths exceeding 20 GHz, suggesting a future need for hybrid analog-digital approaches.Power-Performance Trade-offs: A persistent trade-off exists between performance and power consumption. Low-power designs, exemplified by Dartizio et al. [23] at 380 
μ
W, target IoT applications often utilizing duty-cycled architectures. Conversely, high-bandwidth PLLs, such as Guo et al. [22] (9.05–37.0 GHz), can consume significantly more power, up to 12 mW [30]. This highlights the ongoing need for innovative power management strategies.Comparator and Op-amp-Based Designs: Multi-input comparators [48] generally offer limited SNDR and FoM due to nonlinearity. In contrast, op-amp-based designs [9] can achieve higher SNDR (up to 70 dB) but are typically confined to low speeds (<1 GHz). This suggests that op-amps remain relevant for niche, high-precision applications.Persistent Challenge of Fractional Spurs: Fractional spurs remain a significant challenge. Performance is often capped by natural DAC matching at around −63.7 dBc [39]. While foreground calibration [32] and background methods [14] are commonly used, their power and area overheads are often not fully explored in published literature.Dominance of Foreground and Hybrid Calibration: Foreground calibration techniques are dominant for mismatch correction, demonstrating fast calibration times (e.g., 5.5 
μ
s [32]) and ensuring high SNDR and FoM. The emergence of digital and analog mismatch-shaping techniques provides competitive performance with lower overhead, advocating for the adoption of hybrid calibration strategies in future designs.Versatility Across Applications: PLLs exhibit remarkable versatility across diverse applications, including RF, Bluetooth [36], 5G [34], and even cryogenic interfaces [49]. Fractional-N and ADPLL architectures, further enhanced by multi-core VCOs and DTC, are leading the way in high-performance systems across these varied domains.

**Table 4 sensors-25-05623-t004:** Comparative analysis of published PLL design.

Year	Publication	Tech. (nm)	Area (mm^2^)	Power (mW)	Ref. Freq (MHz)	Output Freq. (GHz)	Phase Noise at 1 MHz Offset (dBc/Hz)	Ref. Spur (dB)	FoM (dB)	Jitter (fs) Integ. Range	Topology
2016	[6] Long Kong	45	0.015	4	22.6	2.4	−113.8	−65	−234.1	0.97 ps	Ring oscillator
2017	[9] C.-W	14	0.257	13.4	26	2.69	−113.6	−87.6	−246	137	TDC/DTC Resolution
2018	[24] D. Cherniak	65	0.42	19.7	52	20.4–24.6	−90	−58	-	-	BBPLL+TPM
	[50] H. Yoon	65	0.95	36.4 (x15 mode)	120	25.0–30.0	−89	−83	-	206 @29.22 (1 KHz–100 MHz)	RFD+GHz-PLL+ ILFMs Using Quadrature
2019	[24] W. Wu	28	0.45	18.9	52x2	6.33	−115.1	−70.2	−249.7	75	Sampling Analog
	[51] J. Seol	28	0.07	3.6	50	2	−120.8	−80	−240.3	0.508 (10 KHz–100 MHz)	OSPLL
2020	[17] Z. Gao	40	0.31	3.48	40	2.56 to 41 (46%)	-	−59	−249.4	182 (10 KHz to 40 MHz)	Type II Fractional N Digital
	[13] T. Seong	65	0.108	9.88	100	5.5 (4.5 to 6.0)	−124.9	−58	−233.8	648 (1 KHz to 300 MHz)	DPLL and mothod TIPM
	[52] M. Mercandelli	28	0.16	18	500	11.9–14.1	-	−73.5	−252.1	58.2	Analog Type _I with Fractional-N
	[29] P. Renukaswamy	28	0.9	11.7	80	8.3 to 11.7	−109.1			314 (10 k to 81 MHz)	QDAC/TPM SS-PLL
2021	[14] H. Park	65	0.146	9.27	100	5.3 (5.2 to 6.0)	−128.8	−77	−239.1	365 (10 KHz to 30 MHz)	DPLL
	[41] W. Wu	14	0.31	14.2/8.2	76.8x2	3.1	-	−72	−250.4/−251.6	80/91.5 (10 KHz to 40 MHz)	Analog Typle II (SPD)
	[53] J. Kim	65	0.21	7.3	150	14 to 16	-	−61	−251	104	Digital SS-PLL
	[31] E. Thaller	16 FinFET	0.5	56	245.76	12.1 to 16.6	−115.13	−75.1	−249.0	47.3/49.9	Digital SS- PLL type
2022	[54] C. Hwang	65	0.139	15.67	100	5.2 (4.4 to 5.4)	−133.4	−64	−242.6	188 (1 KHz to 30 MHz)	DPLL
	[55] S. M. Dartizio	28	0.23	20	250	8.5 to 10	-	−70.2	−251.8	48.6	BBPLL
	[56] X. Geng	65	0.45	14.48	200	24 to 28.2	-	−47	−252.8	60 @ 25.8 GHz (20 KHz to 300 MHz)	CPPLL With TAPFD
	[57] H. Shanan	28	2	187	80 to 200	8.8 to 12	−121		−237.5	97	RTWO_based ADPLL
2023	[18] Y. Jo	65	0.38	9.5	150	3.0 to 3.7	-	−67	−244.9	89	Digital SSPLL
	[39] M. Dartizio	28	0.33	17.2	250	9.25 to 10.5	-	−70.5	−250	76.7	BBPLL with ICS DTC + FCW Subtractive Dithering
	[58] D. Xu	65	0.48	14.2	50	6.5 $\tilde{8}$	-	−72.4	−242.9	191	Frac+Frac
	[59] Q. Zhang	65	0.23	13.56	50	1.5004	−116	−58	−230.6	0.8 ps (10 KHz to 100 MHz)	ADLLs and ILPLL
2024	[60] Y. Shin	40	0.17	15.3	150	10.4 to 11.8	−109.3 @ 100 KHz	−65	−250.5	76 (10 KHz to 100MHz)	Digital SPLL
	[61] M. Rossoni	28	0.21	17.5	250	8.75 to 10.25	-	−69.4	−253.5	57.3 (1KHz–100 MHz)	RCVS-DTC Quatization Error cancellation Approach with DPLL
	[62] A. Narayanan	65	0.54	59-66	106.25	6–12	−108.4 (@1 MHz)	<−60	-	300–510 (1 KHz–40 MHz)	Single PLL with External Analog loop Filter
2025	[21] M. Chae	40	0.12	14.4	100	10 to 11.5	−114.3	-	−253.2	65 (1 KHz to 100 MHz)	Delta sigma Q-error compensation method
	[23] S. M. Dartizio	22	0.3	380 (mW)	24	2.25 to 2.54	−131.2 (@10 MHz)	−71.3	−242.8	1.17 ps	DPLL
	[63] S. Gallucci	28	0.21	9.5	250	4.4 to 5	−146. (@10 MHz)	−80.6	−257	45.8	Digital BB-PLL
	[64] F. Bu	65	0.2	19.2	200	8.8 (7.4 to 9.2)	-	−72.96	−249.9	73.28 (10 KHz–100 MHz)	Analog DSPLL with PI and VI Spur cancellation scheme
	[65] M. Rossoni	28	0.21	17.5	250	8.75–10.25	-	−69.4	−253.5	57.3	BBPLL with RCVS-DTC

**Figure 16 sensors-25-05623-f016:**
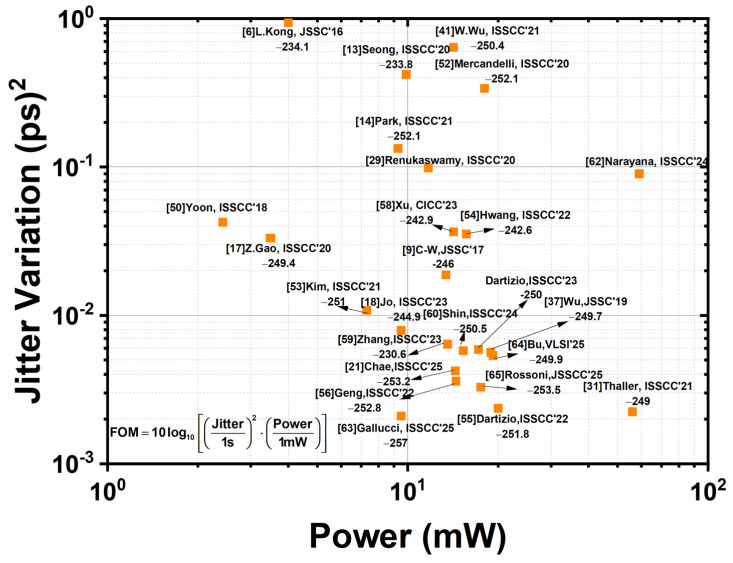
FOMs PLL comparison.

**Figure 17 sensors-25-05623-f017:**
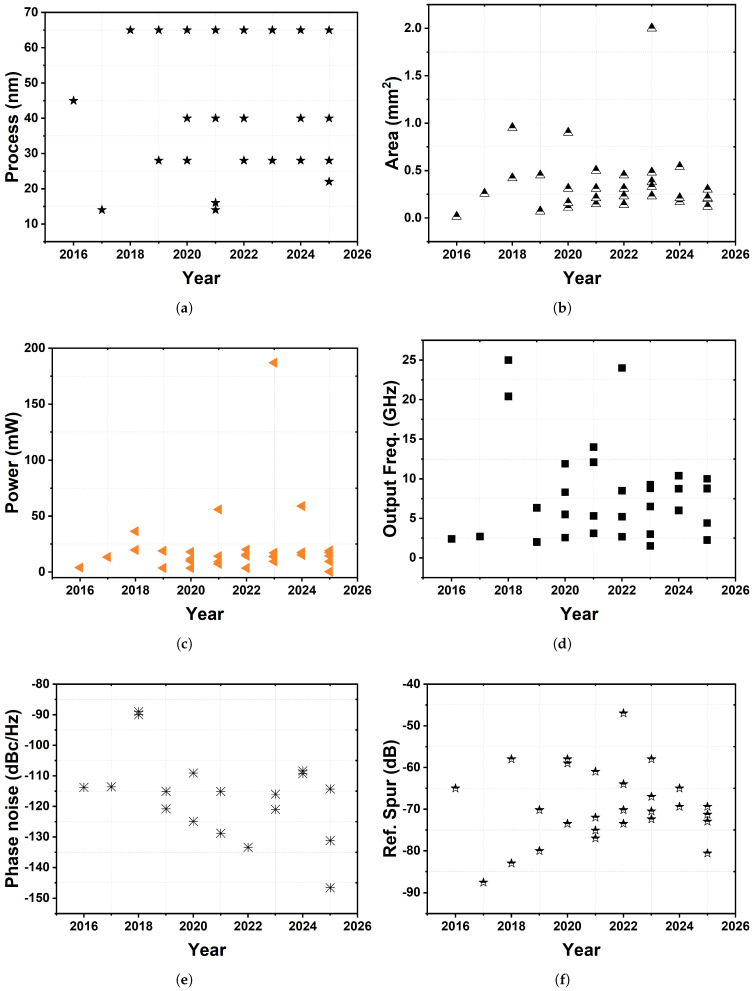
Performance metrics from publications spanning 2016 to 2025: (**a**) Process (nm); (**b**) Area (mm^2^); (**c**) Power (mW); (**d**) Output Frequency (GHz); (**e**) Phase Noise (dBc/Hz); (**f**) Reference Spur (dB).

## 6. Future Trends in PLL Design

As modern electronic systems demand ever-higher performance, lower power consumption, and greater integration, the design of PLL circuits is rapidly evolving to meet these stringent requirements.

### 6.1. Ultra-Low-Power PLLs for IoT and Biomedical Applications

With the explosion of battery-powered and energy-harvesting devices like wearable health monitors, wireless sensor nodes, and implantable biomedical systems, there is a strong need for PLLs that operate at microwatt or even nanowatt power levels. This trend is spurring innovations in subthreshold design techniques, current-reuse architectures, and clockless digitally controlled oscillator (DCO) schemes. A key challenge remains achieving acceptable phase noise and jitter performance under these tight power budgets, demanding clever trade-offs between performance and energy efficiency.

### 6.2. The Rise of Digital and All-Digital PLLs

To address the scalability and reconfigurability needs of modern SoCs, especially in multi-standard wireless transceivers and high-speed data converters, fully digital PLL architectures are gaining prominence. DPLLs offer better compatibility with deep submicron CMOS technologies, improved programmability, and lower sensitivity to PVT variations. Future research is focusing on fast frequency acquisition, low-jitter TDCs, and hybrid analog-digital calibration techniques to overcome the performance bottlenecks traditionally associated with digital designs.

### 6.3. Wideband, Fast-Locking and DDS-Based PLLs for Next-Generation Communications

The emergence of multi-band, multi-protocol communication systems, including 5G/6G, satellite links, and mmWave transceivers, requires PLLs capable of supporting wide tuning ranges while maintaining fast lock times and minimal spurious tones. Frequency synthesizers that can seamlessly adapt across different bands with minimal reconfiguration overhead are a key enabler for future RF front-end architectures. Techniques such as switched-capacitor banks, fractional-N division with sigma-delta modulation, and predictive locking algorithms are expected to play crucial roles in this domain.

Additionally, a promising direction for future growth is the combination of Direct Digital Synthesis (DDS) with Phase-Locked Loop (PLL) architectures, especially in the context of real-time adaptive beamforming [66,67]. For mmWave phased arrays, large MIMO deployments, and software-defined radio platforms, it is necessary to be able to dynamically guide antenna beams by applying accurate phase changes along the local oscillator (LO) route. DDS-PLL hybrids are the only kind of hybrid that can provide ultra-fine frequency and phase resolution, as well as quick configuration and minimal phase noise. This makes them perfect for meeting the strict needs of beamsteering with low latency and high throughput [68,69,70,71].

As wireless systems evolve toward 5G-Advanced and 6G paradigms, the need for reconfigurable, scalable LO generation becomes increasingly critical. DDS-PLL solutions enable per-element LO control in distributed antenna arrays, allowing for more granular and energy-efficient spatial multiplexing. Furthermore, their compatibility with digital control loops enhances integration in CMOS and heterogeneous packaging platforms.

### 6.4. Machine Learning-Assisted and Self-Healing PLLs

As system complexity grows, PLLs will increasingly benefit from the integration of machine learning (ML) techniques for real-time adaptation and performance optimization [72,73]. ML-assisted PLLs could dynamically adjust loop parameters, compensate for aging and mismatch effects, and even predict locking behavior under varying environmental conditions. Early research in this direction shows promise, particularly when combined with digitally intensive control loops and on-chip monitoring circuits.

### 6.5. Robust PLL Design in Advanced CMOS Nodes

As CMOS technology scales below 5 nm, designers must contend with increased leakage currents, reduced voltage headroom, and significant device variability. These factors degrade the analog performance of traditional PLL blocks, such as charge pumps, loop filters, and VCOs. Future designs are expected to rely more heavily on digital calibration, digitally assisted analog blocks, and time-based signal representations to ensure reliable operation. The trend toward “analog in digital disguise” is likely to accelerate, enabling higher levels of integration and portability across technology nodes.

### 6.6. Quantum Optics and Precision Metrology

In addition to the above future PLL design trends, PLL is also an indispensable component in optical frequency comb and pound-drive hall locking techniques. In [74], optical frequency comb (OFC) is a laser light source with a spectrum of thousands of equally spaced frequency lines, so OFC needs to accurately lock at least one frequency to the optical standard, which means OFC needs to use Pound-Drive Hall (PDH) technique, a technique to lock the laser frequency to an optical resonance with extremely high sensitivity and accuracy to obtain errors. Then, PLL will play a role in controlling the laser source or the internal parameters of the comb to reduce jitter and to ensure that the comb is always “synchronized” with time. Thus, PLL ensures phase and frequency stability, so that the comb always reports closely to the frequency standard (optical or RF). In addition, PLL helps the system operate accurately and stably in the long term, especially in measurement, standardization, and telecommunications.

### 6.7. Integration with Emerging Technologies

Finally, PLLs will continue to play a foundational role in interfacing with emerging computing and communication platforms:In quantum computing systems, ultra-low-jitter PLLs are essential for coherent clock distribution.In edge-AI processors, high-speed clock generation with minimal phase noise is crucial for data throughput.In terahertz systems, novel oscillator topologies and frequency multiplication techniques are needed to push PLLs into regimes beyond 300 GHz.

In summary, the future of PLL design lies at the intersection of energy efficiency, digital scalability, intelligent control, and technology-aware robustness. The next generation of PLLs will not only be performance-centric but also adaptive, resilient, and deeply integrated within heterogeneous system-on-chip platforms.

## 7. Conclusions

This paper has offered a comprehensive look at PLL technology, from its foundational innovations to its indispensable role in modern electronics. We have explored the core principles that allow PLLs to deliver precise frequency synthesis, robust signal synchronization, and remarkably low phase noise. These capabilities are vital for applications spanning wireless communication, radar, and automotive electronics.

Our review highlighted the complex challenges posed by next-generation systems, while also showcasing the cutting-edge solutions and recent advancements across various architectures, including integer-N, fractional-N, DPLLs, ADPLLs, and CPPLLs. These ongoing innovations aim to strike a crucial balance between precision, scalability, and power efficiency.

Looking forward, the future of PLL technology is incredibly promising. Continuous advancements will undoubtedly meet emerging demands, ensuring its pivotal contribution to the ongoing evolution of high-performance communication systems.

## Figures and Tables

**Figure 1 sensors-25-05623-f001:**
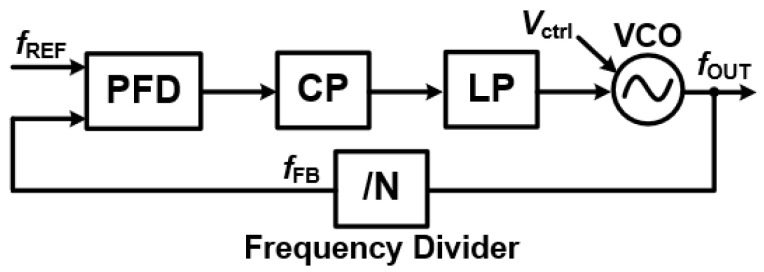
The basic framework of PLL.

**Figure 2 sensors-25-05623-f002:**
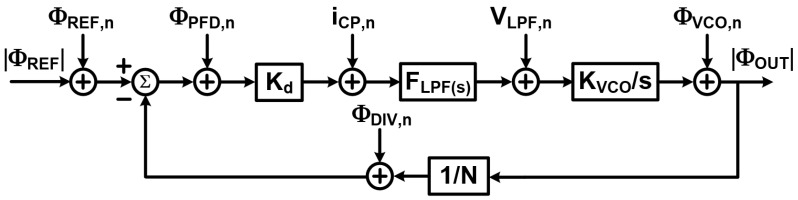
Phase-domain model of PLL.

**Figure 3 sensors-25-05623-f003:**
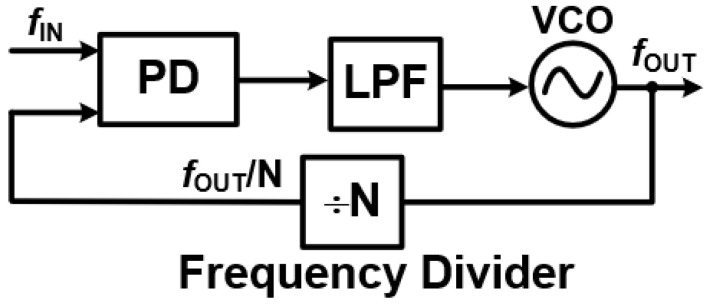
Block diagram of APLL circuit.

**Figure 4 sensors-25-05623-f004:**
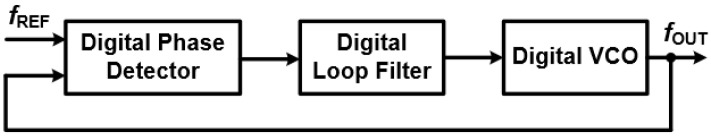
DPLL architecture.

**Figure 5 sensors-25-05623-f005:**
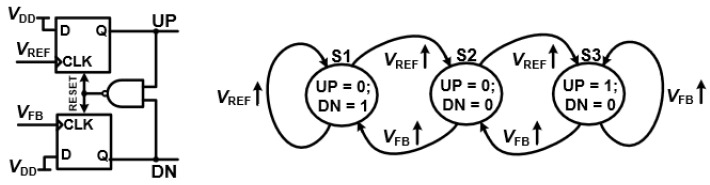
Phase frequency detector of DPLL.

**Figure 6 sensors-25-05623-f006:**
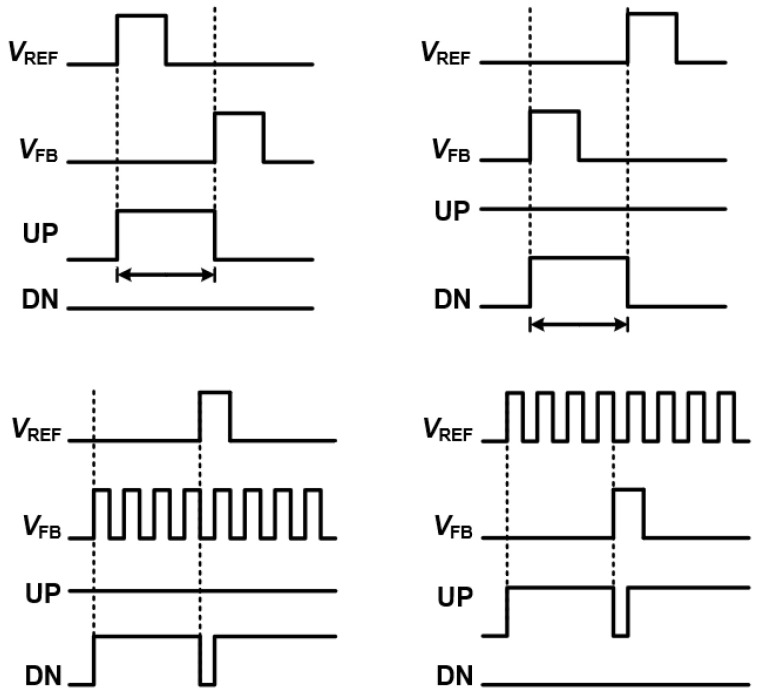
Timing of PFD in DPLL.

**Figure 7 sensors-25-05623-f007:**
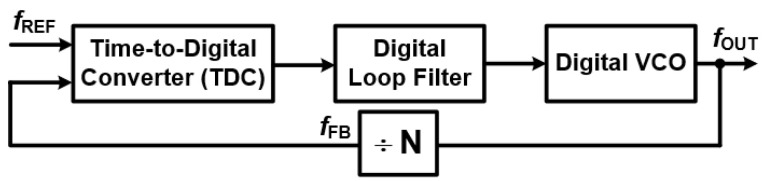
ADPLL architecture.

**Figure 8 sensors-25-05623-f008:**
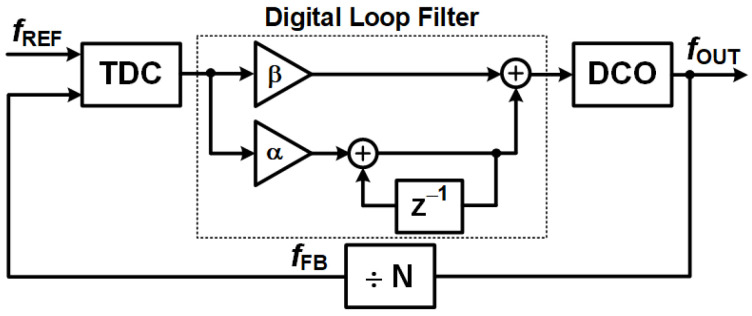
Model of digital loop filter in ADPLL.

**Figure 9 sensors-25-05623-f009:**
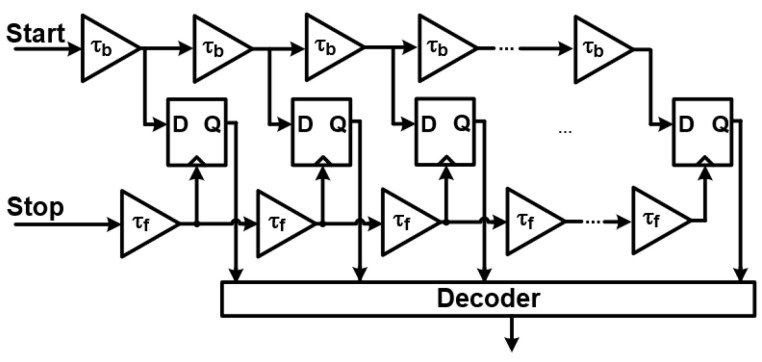
Linear TDC in ADPLL.

**Figure 10 sensors-25-05623-f010:**
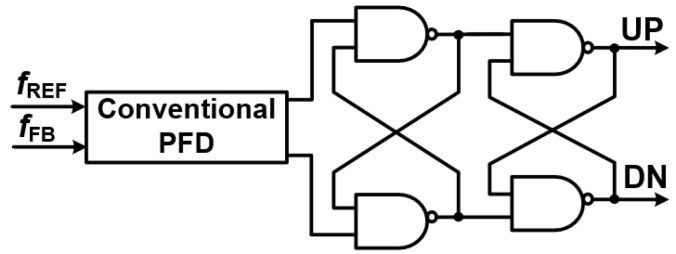
Bang Bang TDC in ADPLL.

**Figure 11 sensors-25-05623-f011:**
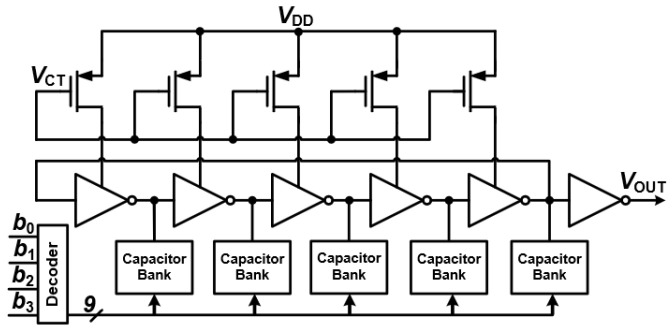
DCO in ADPLL.

**Figure 12 sensors-25-05623-f012:**
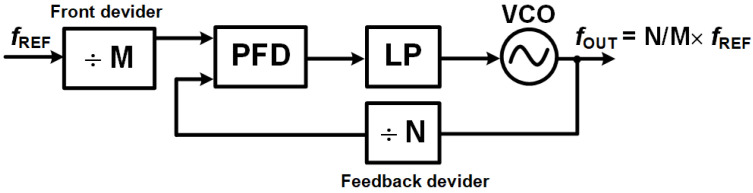
Integer-N PLL architecture.

**Figure 13 sensors-25-05623-f013:**
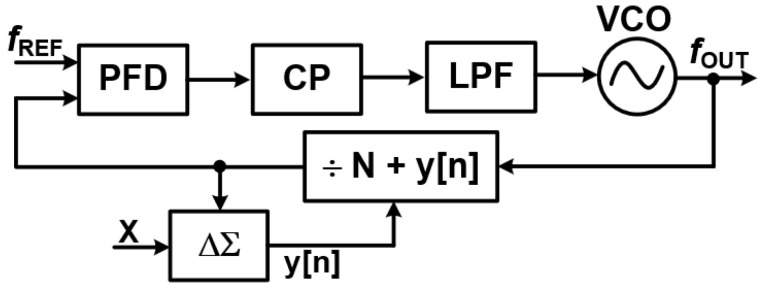
Schematic fractional-N PLL.

**Figure 14 sensors-25-05623-f014:**
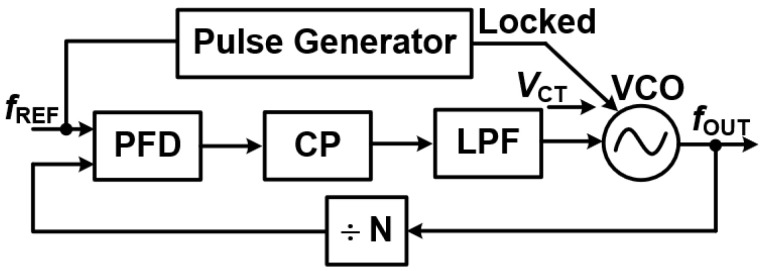
IL PLL architecture.

**Figure 15 sensors-25-05623-f015:**
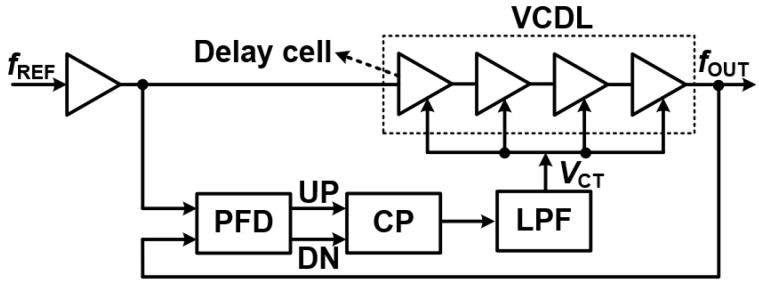
Schematic of DLLs.

**Table 1 sensors-25-05623-t001:** Comparison between charge pump PLL and all-digital PLL.

	Charge Pump PLL	All Digital PLL
Phase Error Information	Pump current	Quantized Digital
Loop Filter	RC filter (passive or active)	Digital filter (FIR or IIR)
Oscillator control	Analog (voltage or current)	Digital code (binary or thermometer)
Noise sensitivity	Higher, due to analog components	Lower, more robust to noise

**Table 2 sensors-25-05623-t002:** Comparative analysis of fractional-N and integer-N Phase-Locked Loops.

Criterion	Fractional-N PLL	Integer-N PLL
Operating Principle	Output frequency is a non-integer multiple of the reference frequency ( $f_{\mathrm{out}}=Nf_{\mathrm{ref}}+\frac{F}{M}f_{\mathrm{ref}}$ ).	Output frequency is an integer multiple of the reference frequency ( $f_{\mathrm{out}}=Nf_{\mathrm{ref}}$ ).
Frequency Resolution	High resolution with fine frequency steps, suitable for agile applications.	Coarse resolution; steps are equal to $f_{\mathrm{ref}}$ .
Complexity	Higher; requires fractional divider and delta-sigma modulator.	Lower; no complex fractional division.
Phase Noise (Jitter)	Elevated due to quantization and modulation noise; requires noise shaping.	Generally lower when referenced to a low-jitter clock.
Spurious Tones	Prone to spurs; needs advanced suppression techniques.	Minimal spurious emissions due to harmonic alignment.
Lock Time	Faster due to finer tuning granularity.	Typically slower for large frequency jumps.
Applications	Wireless communication, SDR, frequency-hopping systems.	Digital clocks, fixed-frequency generation.
Power Consumption	Higher due to additional modulation circuitry.	Lower with simpler architecture.

**Table 3 sensors-25-05623-t003:** Comparison of different PLL architectures.

Criteria	APLL	DPLL	ADPLL	Integer-N PLL	Fractional-N PLL	IL PLL	DLL
Control Type	Analog signal-based	Mixed-signal	Fully digital	Frequency divider is integer	Frequency divider is fractional	Locked via injection	Delay control instead of phase lock
Phase Detector	XOR or analog mixer	Digital PFD	Digital PFD	Digital PFD	Digital PFD	Often not needed or simplified	PFD or simple PFD
Oscillator Type	VCO (analog)	Digital controlled VCO or hybrid	DCO	VCO (analog)	VCO (analog)	Injection-locked VCO	No VCO, use delay line
Loop filter	Analog filter (RC, active)	Digital or mixed filter	Digital filter	Analog filter	Analog or Digital filter	Often minimal or absent	Often minimal or not needed
Phase noise/ Jitter	Potentially low	Design dependent	Good, digital control friendly	Good	Can be higher (due to dithering)	Good within range	Very low (no VCO noise)
Lock time	Moderate to fast	Moderate to fast	Very fast	Depends on N	Slower than Integer-N	Very fast	Fast
Design Complexity	Moderate to high	Higher than analog	High (fully digital design)	Low to moderate	High (requires ΔΣ )	Moderate	Low to moderate
Flexibility	Limited by analog hardware	More flexible than analog	Very flexible	Limited (integer only)	Highly flexible (fractional step)	Limited by injection physics	Limited (depends on delay line)
Applications	RF, analog communications	Telecom, digital electronics	SoC, digital RF, processors	Basic frequency synthesis	Multichannel RF, GSM, LTE	Clock recovery, high-speed clocks	DDR, SDRAM, high-speed data systems

## Data Availability

Data sharing is not applicable to this article.

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
