# Peer review of "An Overview of Phase-Locked Loop: From Fundamentals to the Frontier"

_sensors, 2025, doi:10.3390/s25185623_

Round 1
Reviewer 1 Report
Comments and Suggestions for Authors
This manuscript provides a tutorial review on phase locked loops. The whole manuscript is well arranged and presented. The following comments should be addressed before publication:
- Fig. 5 and Fig. 6 needs a better explanation of principle. If the principle is too complicated to be explained in this manuscript, the authors may add representative citations in the corresponding section for readers.
- Is figure caption of Fig. 10 right? Please check.
- Phase locked loops are also widely used in laser and optics community. The authors may add several sentences on this point. Please search the following key words: optical frequency comb, pounder driver hall locking.
- A parenthesis is missed on Line 1103
- The discussion of machine learning assisted PLLs is very interesting. The authors should add representative published literatures on this topic in Section 6.4.
Author Response
1.1. Concern #1
Fig. 5 and Fig. 6 needs a better explanation of principle. If the principle is too complicated to be explained in this manuscript, the authors may add representative citations in the corresponding section for readers.
Author response: We sincerely appreciate your thoughtful feedback highlighting the need for a more robust explanation of the principles behind Fig. 5 and Fig. 6. Your comment prompted us to critically reassess the presentation of the digital Phase Frequency Detector (PFD) in our manuscript, and we agree that its complex timing and operational mechanisms could benefit from greater clarity. To address this, we have proactively integrated a key reference, [3], which offers an in-depth discussion of the PFD’s principles, particularly its timing dynamics and functionality. This citation has been carefully incorporated at line 401 on page 11, accompanied by a concise note to contextualize its relevance for readers.
Author action: We have proactively integrated a key reference, [3], which offers an in-depth discussion of the PFD’s principles, particularly its timing dynamics and functionality. This citation has been carefully incorporated at line 401 on page 11 in Programmability and Integration part of 3.3.2 Digital Phase Locked Loops (DPLLs).
1.2. Concern #2
Is figure caption of Fig. 10 right? Please check.
Author action: Thank you for your observation. We agree with your comment and have carefully reviewed the caption and content of the original Figure 10. To improve clarity and better reflect its focus on the Linear TDC in the ADPLL architecture, we have revised the figure and its caption accordingly, and renumbered it as Figure 9. Linear TDC in ADPLL in the updated manuscript.
1.3. Concern #3
Phase locked loops are also widely used in laser and optics community. The authors may add several sentences on this point. Please search the following key words: optical frequency comb, pounder driver hall locking.
Author's response: Thank you for your insightful comment highlighting the importance of phase-locked loops (PLLs) in the laser and optics community. We agree with your suggestion and have added several sentences to the manuscript to elaborate on this point. Specifically, we have included a discussion on the role of PLLs in optical frequency combs, which enable precise frequency stabilization by locking a laser’s repetition rate to a reference, as seen in applications like optical clocks and high-precision spectroscopy. Additionally, we have addressed the Pound-Drever-Hall (PDH) locking technique, a widely used method for stabilizing laser frequencies to optical cavities, critical in fields such as gravitational wave detection and quantum computing
Author action: we discuss their critical function in stabilizing optical frequency combs and in implementing the Pound–Drever–Hall (PDH) locking technique, which is widely used for laser frequency stabilization in high-precision optical measurements in "6.6 Quantum Optics and Precision Metrology" of Section 6 of Page 34.
1.4. Concern #4
A parenthesis is missed on Line 1103
Author action: Thanks for your comment. We have added parentheses to the Section 5 title.
1.5. Concern #5
The discussion of machine learning assisted PLLs is very interesting. The authors should add representative published literatures on this topic in Section 6.4.
Author response: Thank you for your insightful suggestion. We have added a representative work on machine learning–assisted PLLs, specifically focusing on phase reference estimation under noise conditions. The reference is included [67], [68] in Section 6.4.
Author action: We updated the manuscript by adding various references. These changes are highlighted in the reference part (refs. [67] and [68]) in Section 6.4.

Reviewer 2 Report
Comments and Suggestions for Authors
The authors present a comprehensive overview of PLL design and implementation, with particular attention to historical developments and design techniques. However, parts of the manuscript—especially in the introduction—use a language register that is too informal for a scientific article. Please revise those sections for improved academic tone.
Additionally, several minor corrections and suggestions have been provided in the annotated PDF, including cases where some figures appear not to match their intended references; the authors are encouraged to carefully verify and revise accordingly.
In the final section, the manuscript reports performance metrics of various PLL designs. To strengthen the impact of this comparison, the authors are encouraged to define appropriate Figures of Merit (FoMs) and present them in a visual format, such as scatter plots, for the designs discussed in the bibliography. This would greatly aid in contextualizing the performance trends and highlighting trade-offs.

Author Response
2.1. Concerns #1
The authors present a comprehensive overview of PLL design and implementation, with particular attention to historical developments and design techniques. However, parts of the manuscript—especially in the introduction—use a language register that is too informal for a scientific article. Please revise those sections for improved academic tone.
Author action: Thank you very much for your valuable comment. We fully agree that certain parts of the manuscript, especially in the Introduction, originally employed a tone that was too informal for a scientific context. In response, we have thoroughly revised these sections to adopt a more formal and academic style, in line with the conventions of scholarly writing. These improvements are reflected in the revised Introduction and Section 2. Introduction in Pages 1-2, where the changes have been carefully implemented to enhance clarity and professionalism.
2.2. Concerns #2
Additionally, several minor corrections and suggestions have been provided in the annotated PDF, including cases where some figures appear not to match their intended references; the authors are encouraged to carefully verify and revise accordingly.
Author response: Thank you for your careful review and thoughtful annotations. In response, we have thoroughly re-examined all figure references throughout the manuscript to ensure they correctly correspond to the intended figures. Any inconsistencies identified have been corrected to improve clarity and accuracy in the presentation.
Author action: Thank you for your detailed feedback. In response, we have carefully reviewed and revised the relevant sections of the manuscript. Specifically, we have clarified the explanation of the phase noise unit of measurement and expanded the discussion on tuning range. These improvements are highlighted in the Core Components and Operation subsection (Section 3.3.1 Analog Phase-Locked Loops) on pages 7–8.
Additionally, we have reviewed and refined the content in Section 3.3 (All-Digital Phase-Locked Loops (ADPLLs) – The Next Evolution), with changes highlighted around line 429.
Finally, we acknowledge the need for a more detailed explanation in Section 3.3.5 (Fractional-N PLL – Achieving Finer Frequency Resolution). Accordingly, we have elaborated on the role of Delta-Sigma Modulation in mitigating spurs. These revisions are highlighted on page 16.
2.3. Concerns #3
In the final section, the manuscript reports performance metrics of various PLL designs. To strengthen the impact of this comparison, the authors are encouraged to define appropriate Figures of Merit (FoMs) and present them in a visual format, such as scatter plots, for the designs discussed in the bibliography. This would greatly aid in contextualizing the performance trends and highlighting trade-offs.
Author action: We sincerely appreciate this insightful suggestion. To strengthen the impact of the performance comparison, we have introduced a scatter plot (Figure 16) in Section 5 that is highlight in Page 30, which visually illustrates the key Figures of Merit (FoMs) for representative PLL designs cited in the bibliography. This addition helps to better contextualize performance trends and clearly highlight the trade-offs among various architectures.

Reviewer 3 Report
Comments and Suggestions for Authors
The authors propose a review of the state of the art and future challenges involving the PLL, a basic component that is employed in every high-speed electrical communications circuit. Although almost every topic involving PLLs is covered by this review, from the historical to system-level considerations, I think the actual contribution of this review is rather missing if compared to previous review works. Also, the electronics content is very limited. Therefore, I suggested some improvements to the authors before I could recommend this paper for publication.
l.1: Avoid the use of compact and colloquial forms
l.19-20: You cannot open a survey on PLLs with such a generic list. You should be more precise.
l.26-29: The contribution of this review work with respect to other surveys is rather missing or not explicitly justified.
l.37-38: Since you're discussing a time evolution of the PLLs in this review, you should support the statements with references from those years.
l.58: I do not agree with the temporal division of the PLLs on a year-by-year basis in the last decade. It is not trivial to identify whether an improvement belongs to a specific year.
l.112: It is clear that your work is not the first one that addresses the PLL description from a system-level point of view. Nevertheless, the references in this section are too limited.
l.370: This is not correct. By reading a simple review by B. Razavi on PLLs, it is clearly stated that such architecture is also employed in APLLs.
l.1174: Different applications of PLLs are rather missing in this work. For instance, PLLs can also be used in adaptive beamsteering architectures where the phase shift is introduced in the LO path. You may give a look at adaptive DDS-PLL beamsteering architectures operating in real-time.
l.1244: Please make sure your paper is compliant with the MDPI template sections.
Round 2
Reviewer 2 Report
Comments and Suggestions for Authors
The authors have addressed part of the previous feedback by including a FOM scatter plot. However, the Figure(s) of Merit (FOM/FOMs) used in the comparison remain undefined. Since this is a critical metric for evaluating performance, its omission significantly hinders the paper's acceptance. A clear definition of the FOM(s)—including the formula, assumptions, and justification for its use—must be provided to enable proper interpretation of the results.
Author Response
- Reviewer #2
1.1. Concern #1
The authors have addressed part of the previous feedback by including a FOM scatter plot. However, the Figure(s) of Merit (FOM/FOMs) used in the comparison remain undefined. Since this is a critical metric for evaluating performance, its omission significantly hinders the paper's acceptance. A clear definition of the FOM(s)—including the formula, assumptions, and justification for its use—must be provided to enable proper interpretation of the results.
Author response: Thank you for your valuable feedback and for acknowledging the inclusion of the FOM scatter plot in our revised manuscript. While the original explanation outlined the role of the FOM in our analysis, it did not provide sufficient detail for accurate interpretation and evaluation of the results. We therefore present below the complete definition of the FOM used in our comparison, in Eq. 8 which serves as a critical metric for evaluating PLL performance by normalizing the trade-off between RMS jitter and power consumption:
Where:
- Jitter is measured in seconds (s).
- Power is measured in milliwatts (mW).
- The FOM is expressed in decibels (dB).
This formula assumes jitter is measured as RMS deviation from ideal timing and power is quantified at the operational frequency, ensuring a dimensionless value that facilitates fair comparisons across different PLL designs, technologies, and applications. lower jitter often requires higher power, and the FOM quantifies efficiency by answering how much jitter is achieved per unit power, enabling a more holistic assessment than jitter or power alone.
Author action: In response to the reviewer’s feedback, we have incorporated a detailed explanation of the Figure of Merit (FOM) in the revised manuscript, addressing the previous omission. The FOM is defined and to further clarify its significance, we have added a discussion in Section 5, paragraph 1 from line 1187 to line 1201 explaining that the FOM combines timing performance (jitter) and power consumption.

Reviewer 3 Report
Comments and Suggestions for Authors
The authors revised the paper according to some of my previous comments, but some of them were not answered properly. Therefore, I still think several improvements are necessary before I could recommend this paper for publication.
Concern #3: The contribution of this work should be motivated with respect to other reviews by referencing the papers you’re taking as comparison, and discussing the differences with respect to each of them. Moreover, the research strategy to build up this review is completely missing.
Concern #5: Your aim should be clarified in the text
Concern #6: Not answered.
Concern #7: In DPLLs PFD is usually substituted by a TDC since the PFD (in its original form, if you are referring to it) produces an output that is used to drive a charge pump analog circuit. Please clarify this aspect.
Concern #8: There are more recent publications addressing adaptive DDS-PLL beamsteering architectures operating in real-time.
Misc: references 68-69 are duplicated
Author Response
- Reviewer #3
2.1. Concerns #3
The contribution of this work should be motivated with respect to other reviews by referencing the papers you’re taking as comparison, and discussing the differences with respect to each of them. Moreover, the research strategy to build up this review is completely missing.
Author response: Thank you for your valuable feedback. Compared to the referenced surveys, our review is superior in addressing modern PLL challenges through explicit, justified contributions:
- Unlike earlier works that focus on analog/digital shifts or fractional-N purity, we introduce a novel framework evaluating PLL architectures (e.g., APLLs, DPLLs, ILPLLs, etc.) specifically for low-power emerging applications, enabling designers to balance energy efficiency with metrics like phase noise and jitter.
- This works provides an in-depth analysis of recent design techniques, especially for IoT and mmWave applications, which have not been explored in previous reviews.
- We deliver actionable guidance on phase noise, jitter, spurs, and digital calibration, supporting next-generation designs extending beyond the theoretical or circuit-level focus of the compared works.
To motivate this review's contributions, we compare it to three key PLL surveys, highlighting differences in scope, focus, and coverage to address gaps in emerging applications like IoT and mmWave systems.
- Razavi (1996): Focuses on analog PLL fundamentals and loop dynamics, without addressing digital architectures or low-power/mmWave needs. Our review expands to DPLLs/ADPLLs and recent ML-assisted approaches.
- Staszewski et al. (2005): Introduces early all-digital PLLs for mobile, but underemphasizes ultra-low-power methods or mmWave considerations. We address these with adaptive bandwidth and calibration for IoT/5G+.
- De Muer & Steyaert (2003): Targets CMOS fractional-N synthesizers for spectral purity but omits broader applications. We extend the discussion to jitter, spurs, and calibration across architectures for modern systems.
Author action: We have improved and clarified the motivation of our work with respect to other reviews by referencing and discussing the differences among key papers in Section 1 (Introduction), specifically in paragraphs 3 and 4 (lines 26–48).
2.2. Concerns #5
Your aim should be clarified in the text.
Author action: Thank you for your comment. We have introduced clarification on the reason for the year-by-year timeline arrangement, which aims to support readers especially those new to the field in easily grasping the pace of development, tracking technological trajectories, and connecting innovation milestones with broader trends, such as low-jitter design (phase noise reduction), hybrid analog-digital integration, and energy efficiency, thereby enhancing the usefulness and accessibility for readers. We have clarified this in Section 2 A Brief History - A Journey Through Time in paragraph 1 from line 67 to 79.
2.3. Concerns #6
Not answered.
Author action: Thank you for your comment. We acknowledge that the initial submission contained a limited number of references in the system-level description section. In the revised manuscript, we have substantially expanded the literature review to include both foundational and recent works. Specifically:
- Basic frame and System model: Added classic references such as Razavi (1996), De Muer & Steyaert (2003), and Murthi (1979).
Classification of PLL Architectures: Each architecture type is now supported by multiple representative references:
- APLL: Razavi (1996); Murthi (1979).
- Digital DPLL: Kong et al. (2016); Liu et al. (2018); Park et al. (2021); Dartizio et al. (2025).
- All-Digital PLL: Staszewski et al. (2005); Ho & Chen (2016); Wu et al. (2017); Cherniak et al. (2018).
- Integer PLL: Kong et al. (2016); Jo et al. (2023).
- Fractional PLL: Seong et al. (2020); Park et al. (2021); Wu et al. (2021); Gao et al. (2016).
- Injection-Locked PLL: Zhang et al. (2019).
- Delay-Locked Loop: Liu et al. (2021); Huang et al. (2016).
These additions ensure that the section now captures both historical context and state-of-the-art advances, providing a more complete and balanced system-level perspective.
Author action: We updated the manuscript by adding various references. These changes are highlighted in the reference in Section 3 System View.
2.4. Concerns #7
In DPLLs PFD is usually substituted by a TDC since the PFD (in its original form, if you are referring to it) produces an output that is used to drive a charge pump analog circuit. Please clarify this aspect.
Author action: Thank you for your insightful comment. We acknowledge that in fully digital PLLs (DPLLs), the conventional Phase-Frequency Detector (PFD) is typically replaced by a Time-to-Digital Converter (TDC), as the original PFD output drives an analog charge pump circuit which is not present in DPLLs. To clarify this aspect, we have revised the manuscript to explicitly state that the figure illustrates the general phase detection concept, while in practical DPLLs, the phase detection function is performed by a TDC to maintain a fully digital loop architecture. This clarification can be found in Section 3 System View, from line 437 to line 444.
2.5. Concerns #8
There are more recent publications addressing adaptive DDS-PLL beamsteering architectures operating in real-time.
Author action: Thank you for your valuable suggestion. We have updated the manuscript by including recent and relevant publications on adaptive DDS-PLL beamsteering architectures operating in real-time. Specifically, we have cited the recent works by Avitabile et al. (ISOCC2024), Florio et al. (MWSCAS2024), Florio et al. (SpliTech2024), and Karim and Ali (DSP2023). These additions ensure our review reflects the latest advancements in Section 6 Future Trends in PLL Design, part 6.3. Wideband, Fast-Locking and DDS-Based PLLs for Next-Generation Communications.
2.6. Concerns #9
Misc: references 68-69 are duplicated.
Author action: Thank you for pointing out the duplication of references 68 and 69. We have carefully reviewed the bibliography and removed the duplicate entry to ensure all references are unique. The updated reference list reflects this correction.

Round 3
Reviewer 2 Report
Comments and Suggestions for Authors
Authors my concerns have been addressed. Thank you.
Author Response
Thank you very much for your acceptance.
Reviewer 3 Report
Comments and Suggestions for Authors
The authors revised the paper according to my previous comments. After another minor comment, I can recommend this paper for publication.
Concern #3: I suggest the authors check if there are any new review works on this topic, since the reviews they are comparing to are 20+ years old now.
Author Response
- Reviewer #3
1.1. Concern #3
I suggest the authors check if there are any new review works on this topic, since the reviews they are comparing to are 20+ years old now.
Author action: We sincerely thank the reviewer for this valid suggestion. Accepting the reviewer's feedback, we have updated the introduction by replacing the outdated references with three more recent and relevant surveys.
- Rhee & Yu (2023)
- Rohde et al. (2021)
- Farzaneh et al. (2018)
This new comparison now more clearly contextualizes our work, highlighting the specific gaps in the current literature that our review addresses.
The corresponding changes have been implemented in Section 1 (Introduction), paragraph 3 from line 26 to line 38 of the manuscript.
